# Random Feature Stein Discrepancies

**Jonathan H. Huggins**[*]
Department of Biostatistics, Harvard
jhuggins@mit.edu

**Lester Mackey**[*]
Microsoft Research New England
lmackey@microsoft.com

## Abstract

Computable Stein discrepancies have been deployed for a variety of applications, ranging from sampler selection in posterior inference to approximate Bayesian inference to goodness-of-fit testing. Existing convergence-determining Stein discrepancies admit strong theoretical guarantees but suffer from a computational cost that grows quadratically in the sample size. While linear-time Stein discrepancies have been proposed for goodness-of-fit testing, they exhibit avoidable degradations in testing power—even when power is explicitly optimized. To address these shortcomings, we introduce *feature Stein discrepancies* ($\Phi$SDs), a new family of quality measures that can be cheaply approximated using importance sampling. We show how to construct $\Phi$SDs that provably determine the convergence of a sample to its target and develop high-accuracy approximations—*random $\Phi$SDs* (R$\Phi$SDs)—which are computable in near-linear time. In our experiments with sampler selection for approximate posterior inference and goodness-of-fit testing, R$\Phi$SDs perform as well or better than quadratic-time KSDs while being orders of magnitude faster to compute.

## 1 Introduction

Motivated by the intractable integration problems arising from Bayesian inference and maximum likelihood estimation [9], Gorham and Mackey [10] introduced the notion of a Stein discrepancy as a quality measure that can potentially be computed even when direct integration under the distribution of interest is unavailable. Two classes of computable Stein discrepancies—the graph Stein discrepancy [10, 12] and the kernel Stein discrepancy (KSD) [7, 11, 19, 21]—have since been developed to assess and tune Markov chain Monte Carlo samplers, test goodness-of-fit, train generative adversarial networks and variational autoencoders, and more [7, 10–12, 16–19, 27]. However, in practice, the cost of these Stein discrepancies grows quadratically in the size of the sample being evaluated, limiting scalability. Jitkrittum et al. [16] introduced a special form of KSD termed the finite-set Stein discrepancy (FSSD) to test goodness-of-fit in linear time. However, even after an optimization-based preprocessing step to improve power, the proposed FSSD experiences a unnecessary degradation of power relative to quadratic-time tests in higher dimensions.

To address the distinct shortcomings of existing linear- and quadratic-time Stein discrepancies, we introduce a new class of Stein discrepancies we call *feature Stein discrepancies* ($\Phi$SDs). We show how to construct $\Phi$SDs that provably determine the convergence of a sample to its target, thus making them attractive for goodness-of-fit testing, measuring sample quality, and other applications. We then introduce a fast importance sampling-based approximation we call *random $\Phi$SDs* (R$\Phi$SDs). We provide conditions under which, with an appropriate choice of proposal distribution, an R$\Phi$SD is close in relative error to the $\Phi$SD with high probability. Using an R$\Phi$SD, we show how, for any $\gamma > 0$, we can compute $O_P(N^{-1/2})$-precision estimates of an $\Phi$SD in $O(N^{1+\gamma})$ (near-linear) time when the $\Phi$SD precision is $\Omega(N^{-1/2})$. Additionally, to enable applications to goodness-of-fit testing,

---

[*]Contributed equally

we (1) show how to construct RΦSDs that can distinguish between arbitrary distributions and (2) describe the asymptotic null distribution when sample points are generated i.i.d. from an unknown distribution. In our experiments with biased Markov chain Monte Carlo (MCMC) hyperparameter selection and fast goodness-of-fit testing, we obtain high-quality results—which are comparable to or better than those produced by quadratic-time KSDs—using only ten features and requiring orders-of-magnitude less computation.

**Notation** For measures $\mu_1, \mu_2$ on $\mathbb{R}^D$ and functions $f : \mathbb{R}^D \to \mathbb{C}$, $k : \mathbb{R}^D \times \mathbb{R}^D \to \mathbb{C}$, we let $\mu_1(f) := \int f(x)\mu_1(\mathrm{d}x)$, $(\mu_1 k)(x') := \int k(x, x')\mu_1(\mathrm{d}x)$, and $(\mu_1 \times \mu_2)(k) := \int \int k(x_1, x_2)\mu_1(\mathrm{d}x_1)\mu_2(\mathrm{d}x_2)$. We denote the generalized Fourier transform of $f$ by $\hat{f}$ or $\mathscr{F}(f)$ and its inverse by $\mathscr{F}^{-1}(f)$. For $r \geq 1$, let $L^r := \{f : \|f\|_{L^r} := (\int |f(x)|^r \, \mathrm{d}x)^{1/r} < \infty\}$ and $C^n$ denote the space of $n$-times continuously differentiable functions. We let $\overset{\mathcal{D}}{\Longrightarrow}$ and $\overset{P}{\to}$ denote convergence in distribution and in probability, respectively. We let $\overline{a}$ denote the complex conjugate of $a$. For $D \in \mathbb{N}$, define $[D] := \{1, \ldots, D\}$. The symbol $\gtrsim$ indicates greater than up to a universal constant.

## 2 Feature Stein discrepancies

When exact integration under a target distribution $P$ is infeasible, one often appeals to a discrete measure $Q_N = \frac{1}{N}\sum_{n=1}^{N}\delta_{x_n}$ to approximate expectations, where the sample points $x_1, \ldots, x_N \in \mathbb{R}^D$ are generated from a Markov chain or quadrature rule. The aim in sample quality measurement is to quantify how well $Q_N$ approximates the target in a manner that (a) recognizes when a sample sequence is converging to the target, (b) highlights when a sample sequence is not converging to the target, and (c) is computationally efficient. It is natural to frame this comparison in terms of an integral probability metric (IPM) [20], $d_{\mathcal{H}}(Q_N, P) := \sup_{h \in \mathcal{H}} |Q_N(h) - P(h)|$, measuring the maximum discrepancy between target and sample expectations over a class of test functions. However, when generic integration under $P$ is intractable, standard IPMs like the 1-Wasserstein distance and Dudley metric may not be efficiently computable.

To address this need, Gorham and Mackey [10] introduced the Stein discrepancy framework for generating IPM-type quality measures with no explicit integration under $P$. For any approximating probability measure $\mu$, each Stein discrepancy takes the form

$$d_{\mathcal{TG}}(\mu, P) = \sup_{g \in \mathcal{G}} |\mu(\mathcal{T}g)| \quad \text{where} \quad \forall g \in \mathcal{G}, P(\mathcal{T}g) = 0.$$

Here, $\mathcal{T}$ is an operator that generates mean-zero functions under $P$, and $\mathcal{G}$ is the *Stein set* of functions on which $\mathcal{T}$ operates. For concreteness, we will assume that $P$ has $C^1$ density $p$ with support $\mathbb{R}^d$ and restrict our attention to the popular *Langevin Stein operator* [10, 21] defined by $\mathcal{T}g := \sum_{d=1}^{D} \mathcal{T}_d g_d$ for $(\mathcal{T}_d g_d)(x) := p(x)^{-1}\partial_{x_d}(p(x)g_d(x))$ and $g : \mathbb{R}^D \to \mathbb{R}^D$. To date, two classes of computable Stein discrepancies with strong convergence-determining guarantees have been identified. The graph Stein discrepancies [10, 12] impose smoothness constraints on the functions $g$ and are computed by solving a linear program, while the kernel Stein discrepancies [7, 11, 19, 21] define $\mathcal{G}$ as the unit ball of a reproducing kernel Hilbert space and are computed in closed-form. Both classes, however, suffer from a computational cost that grows quadratically in the number of sample points. Our aim is to develop alternative discrepancy measures that retain the theoretical and practical benefits of existing Stein discrepancies at a greatly reduced computational cost.

Our strategy is to identify a family of convergence-determining discrepancy measures that can be accurately and inexpensively approximated with random sampling. To this end, we define a new domain for the Stein operator centered around a *feature function* $\Phi : \mathbb{R}^D \times \mathbb{R}^D \to \mathbb{C}$ which, for some $r \in [1, \infty)$ and all $x, z \in \mathbb{R}^D$, satisfies $\Phi(x, \cdot) \in L^r$ and $\Phi(\cdot, z) \in C^1$:

$$\mathcal{G}_{\Phi,r} := \left\{ g : \mathbb{R}^D \to \mathbb{R} \mid g_d(x) = \int \Phi(x, z)\overline{f_d(z)}\,\mathrm{d}z \quad \text{with} \quad \sum_{d=1}^{D}\|f_d\|_{L^s}^2 \leq 1 \text{ for } s = \frac{r}{r-1} \right\}.$$

When combined with the Langevin Stein operator $\mathcal{T}$, this *feature Stein set* gives rise to a *feature Stein discrepancy* (ΦSD) with an appealing explicit form $(\sum_{d=1}^{D}\|\mu(\mathcal{T}_d\Phi)\|_{L^r}^2)^{1/2}$:

$$\Phi\mathrm{SD}_{\Phi,r}^2(\mu, P) := \sup_{g \in \mathcal{G}_{\Phi,r}} |\mu(\mathcal{T}g)|^2 = \sup_{g \in \mathcal{G}_{\Phi,r}} \left| \sum_{d=1}^{D} \mu(\mathcal{T}_d g_d) \right|^2$$

$$= \sup_{f:v_d=\|f_d\|_{L^s}, \|v\|_2 \leq 1} \left| \sum_{d=1}^{D} \int \mu(\mathcal{T}_d \Phi)(z) \overline{f_d(z)} \, dz \right|^2$$

$$= \sup_{v:\|v\|_2 \leq 1} \left| \sum_{d=1}^{D} \|\mu(\mathcal{T}_d \Phi)\|_{L^r} v_d \right|^2 = \sum_{d=1}^{D} \|\mu(\mathcal{T}_d \Phi)\|_{L^r}^2. \tag{1}$$

In Section 3.1, we will show how to select the feature function $\Phi$ and order $r$ so that $\Phi\mathrm{SD}_{\Phi,r}$ provably determines convergence, in line with our desiderata (a) and (b).

To achieve efficient computation, we will approximate the $\Phi\mathrm{SD}$ in expression (1) using an importance sample of size $M$ drawn from a proposal distribution with (Lebesgue) density $\nu$. We call the resulting stochastic discrepancy measure a *random* $\Phi\mathrm{SD}$ (R$\Phi\mathrm{SD}$):

$$\mathrm{R}\Phi\mathrm{SD}^2_{\Phi,r,\nu,M}(\mu, P) := \sum_{d=1}^{D} \left( \frac{1}{M} \sum_{m=1}^{M} \nu(Z_m)^{-1} |\mu(\mathcal{T}_d \Phi)(Z_m)|^r \right)^{2/r} \text{ for } Z_1, \ldots, Z_M \overset{\text{i.i.d.}}{\sim} \nu.$$

Importantly, when $\mu$ is the sample approximation $Q_N$, the R$\Phi\mathrm{SD}$ can be computed in $O(MN)$ time by evaluating the $MND$ rescaled random features, $(\mathcal{T}_d \Phi)(x_n, Z_m)/\nu(Z_m)^{1/r}$; this computation is also straightforwardly parallelized. In Section 3.2, we will show how to choose $\nu$ so that $\mathrm{R}\Phi\mathrm{SD}_{\Phi,r,\nu,M}$ approximates $\Phi\mathrm{SD}_{\Phi,r}$ with small relative error.

**Special cases** When $r = 2$, the $\Phi\mathrm{SD}$ is an instance of a kernel Stein discrepancy (KSD) with base reproducing kernel $k(x,y) = \int \Phi(x,z)\overline{\Phi(y,z)} \, dz$. This follows from the definition [7, 11, 19, 21] $\mathrm{KSD}_k(\mu,P)^2 := \sum_{d=1}^{D}(\mu \times \mu)((\mathcal{T}_d \otimes \mathcal{T}_d)k) = \sum_{d=1}^{D} \|\mu(\mathcal{T}_d \Phi)\|_{L^2}^2 = \Phi\mathrm{SD}_{\Phi,2}(\mu,P)^2$. However, we will see in Sections 3 and 5 that there are significant theoretical and practical benefits to using $\Phi\mathrm{SD}$s with $r \neq 2$. Namely, we will be able to approximate $\Phi\mathrm{SD}_{\Phi,r}$ with $r \neq 2$ more effectively with a smaller sampling budget. If $\Phi(x,z) = e^{-i\langle z,x\rangle}\hat{\Psi}(z)^{1/2}$ and $\nu \propto \hat{\Psi}$ for $\Psi \in L^2$, then $\mathrm{R}\Phi\mathrm{SD}_{\Phi,2,\nu,M}$ is the random Fourier feature (RFF) approximation [22] to $\mathrm{KSD}_k$ with $k(x,y) = \Psi(x-y)$. Chwialkowski et al. [6, Prop. 1] showed that the RFF approximation can be a undesirable choice of discrepancy measure, as there exist uncountably many pairs of distinct distributions that, with high probability, cannot be distinguished by the RFF approximation. Following Chwialkowski et al. [6] and Jitkrittum et al. [16], we show how to select $\Phi$ and $\nu$ to avoid this property in Section 4. The random finite set Stein discrepancy [FSSD-rand, 16] with proposal $\nu$ is an $\mathrm{R}\Phi\mathrm{SD}_{\Phi,2,\nu,M}$ with $\Phi(x,z) = f(x,z)\nu(z)^{1/2}$ for $f$ a real analytic and $C_0$-universal [4, Def. 4.1] reproducing kernel. In Section 3.1, we will see that features $\Phi$ of a different form give rise to strong convergence-determining properties.

# 3 Selecting a Random Feature Stein Discrepancy

In this section, we provide guidance for selecting the components of an R$\Phi\mathrm{SD}$ to achieve our theoretical and computational goals. We first discuss the choice of the feature function $\Phi$ and order $r$ and then turn our attention to the proposal distribution $\nu$. Finally, we detail two practical choices of R$\Phi\mathrm{SD}$ that will be used in our experiments. To ease notation, we will present theoretical guarantees in terms of the sample measure $Q_N$, but all results continue to hold if any approximating probability measure $\mu$ is substituted for $Q_N$.

## 3.1 Selecting a feature function $\Phi$

A principal concern in selecting a feature function is ensuring that the $\Phi\mathrm{SD}$ detects non-convergence— that is, $Q_N \overset{\mathcal{D}}{\Longrightarrow} P$ whenever $\Phi\mathrm{SD}_{\Phi,r}(Q_N, P) \to 0$. To ensure this, we will construct $\Phi\mathrm{SD}$s that upper bound a reference KSD known to detect non-convergence. This is enabled by the following inequality proved in Appendix A.

**Proposition 3.1** (KSD-$\Phi\mathrm{SD}$ inequality). *If $k(x,y) = \int \mathscr{F}(\Phi(x,\cdot))(\omega)\overline{\mathscr{F}(\Phi(y,\cdot))(\omega)}\rho(\omega) \, d\omega$, $r \in [1,2]$, and $\rho \in L^t$ for $t = r/(2-r)$, then*

$$\mathrm{KSD}_k^2(Q_N, P) \leq \|\rho\|_{L^t} \Phi\mathrm{SD}_{\Phi,r}^2(Q_N, P). \tag{2}$$

Our strategy is to first pick a KSD that detects non-convergence and then choose $\Phi$ and $r$ such that (2) applies. Unfortunately, KSDs based on many common base kernels, like the Gaussian and Matérn, fail to detect non-convergence when $D > 2$ [11, Thm. 6]. A notable exception is the KSD with inverse multiquadric (IMQ) base kernel.

**Example 3.1** (IMQ kernel). The IMQ kernel is given by $\Psi_{c,\beta}^{\mathrm{IMQ}}(x-y) := (c^2 + \|x-y\|_2^2)^\beta$, where $c > 0$ and $\beta < 0$. Gorham and Mackey [11, Thm. 8] proved that when $\beta \in (-1, 0)$, KSDs with an IMQ base kernel determine weak convergence on $\mathbb{R}^D$ whenever $P \in \mathcal{P}$, the set of distantly dissipative distributions for which $\nabla \log p$ is Lipschitz.[2]

Let $m_N := \mathbb{E}_{X \sim Q_N}[X]$ denote the mean of $Q_N$. We would like to consider a broader class of base kernels, the form of which we summarize in the following assumption:

**Assumption A.** The base kernel has the form $k(x, y) = A_N(x)\Psi(x-y)A_N(y)$ for $\Psi \in C^2$, $A \in C^1$, and $A_N(x) := A(x - m_N)$, where $A > 0$ and $\nabla \log A$ is bounded and Lipschitz.

The IMQ kernel falls within the class defined by Assumption A (let $A = 1$ and $\Psi = \Psi_{c,\beta}^{\mathrm{IMQ}}$). On the other hand, our next result, proved in Appendix B, shows that *tilted base kernels* with $A$ increasing sufficiently quickly also control convergence.

**Theorem 3.2** (Tilted KSDs detect non-convergence). *Suppose that $P \in \mathcal{P}$, Assumption A holds, $1/A \in L^2$, and $H(u) := \sup_{\omega \in \mathbb{R}^D} e^{-\|\omega\|_2^2/(2u^2)}/\hat{\Psi}(\omega)$ is finite for all $u > 0$. Then for any sequence of probability measures $(\mu_N)_{N=1}^\infty$, if $\mathrm{KSD}_k(\mu_N, P) \to 0$ then $\mu_N \overset{\mathcal{D}}{\Longrightarrow} P$.*

**Example 3.2** (Tilted hyperbolic secant kernel). The hyperbolic secant (sech) function is $\mathrm{sech}(u) := 2/(e^u + e^{-u})$. For $x \in \mathbb{R}^D$ and $a > 0$, define the *sech kernel* $\Psi_a^{\mathrm{sech}}(x) := \prod_{d=1}^D \mathrm{sech}(\sqrt{\frac{\pi}{2}}ax_d)$. Since $\hat{\Psi}_a^{\mathrm{sech}}(\omega) = \Psi_{1/a}^{\mathrm{sech}}(\omega)/a^D$, $\mathrm{KSD}_k$ from Theorem 3.2 detects non-convergence when $\Psi = \Psi_a^{\mathrm{sech}}$ and $A^{-1} \in L^2$. Valid tilting functions include $A(x) = \prod_{d=1}^D e^{c\sqrt{1+x_d^2}}$ for any $c > 0$ and $A(x) = (c^2 + \|x\|_2^2)^b$ for any $b > D/4$ (to ensure $A^{-1} \in L^2$).

With our appropriate reference KSDs in hand, we will now design upper bounding $\Phi$SDs. To accomplish this we will have $\Phi$ mimic the form of the base kernels in Assumption A:

**Assumption B.** Assumption A holds and $\Phi(x, z) = A_N(x)F(x-z)$, where $F \in C^1$ is positive, and there exist a norm $\|\cdot\|$ and constants $s, C > 0$ such that

$$|\partial_{x_d} \log F(x)| \leq C(1 + \|x\|^s), \quad \lim_{\|x\| \to \infty}(1 + \|x\|^s)F(x) = 0, \quad \text{and} \quad F(x-z) \leq CF(z)/F(x).$$

In addition, there exist a constant $\underline{c} \in (0, 1]$ and continuous, non-increasing function $f$ such that $\underline{c}\,f(\|x\|) \leq F(x) \leq f(\|x\|)$.

Assumption B requires a minimal amount of regularity from $F$, essentially that $F$ be sufficiently smooth and behave as if it is a function only of the norm of its argument. A conceptually straightforward choice would be to set $F = \mathscr{F}^{-1}(\hat{\Psi}^{1/2})$—that is, to be the *square root kernel* of $\Psi$. We would then have that $\Psi(x-y) = \int F(x-z)F(y-z)\,\mathrm{d}z$, so in particular $\Phi\mathrm{SD}_{\Phi,2} = \mathrm{KSD}_k$. Since the exact square-root kernel of a base kernel can be difficult to compute in practice, we require only that $F$ be a suitable approximation to the square root kernel of $\Psi$:

**Assumption C.** Assumption B holds, and there exists a *smoothness parameter* $\overline{\lambda} \in (1/2, 1]$ such that if $\lambda \in (1/2, \overline{\lambda})$, then $\hat{F}/\hat{\Psi}^{\lambda/2} \in L^2$.

Requiring that $\hat{F}/\hat{\Psi}^{\lambda/2} \in L^2$ is equivalent to requiring that $F$ belongs to the reproducing kernel Hilbert space $\mathcal{K}_\lambda$ induced by the kernel $\mathscr{F}^{-1}(\hat{\Psi}^\lambda)$. The smoothness of the functions in $\mathcal{K}_\lambda$ increases as $\lambda$ increases. Hence $\overline{\lambda}$ quantifies the smoothness of $F$ relative to $\Psi$.

Finally, we would like an assurance that the $\Phi$SD detects convergence—that is, $\Phi\mathrm{SD}_{\Phi,r}(Q_N, P) \to 0$ whenever $Q_N$ converges to $P$ in a suitable metric. The following result, proved in Appendix C, provides such a guarantee for both the $\Phi$SD and the R$\Phi$SD.

**Proposition 3.3.** *Suppose Assumption B holds with $F \in L^r$, $1/A$ bounded, $x \mapsto x/A(x)$ Lipschitz, and $\mathbb{E}_P[A(Z)\|Z\|_2^2] < \infty$. If the* tilted Wasserstein distance

$$\mathcal{W}_{A_N}(Q_N, P) := \sup_{h \in \mathcal{H}} |Q_N(A_N h) - P(A_N h)| \quad (\mathcal{H} := \{h : \|\nabla h(x)\|_2 \leq 1, \forall x \in \mathbb{R}^D\})$$

*converges to zero, then* $\Phi\mathrm{SD}_{\Phi,r}(Q_N, P) \to 0$ *and* $\mathrm{R}\Phi\mathrm{SD}_{\Phi,r,\nu_N,M_N}(Q_N, P) \xrightarrow{P} 0$ *for any choices of* $r \in [1,2]$, $\nu_N$, *and* $M_N \geq 1$.

*Remark* 3.4. When $A$ is constant, $\mathcal{W}_{A_N}$ is the familiar 1-Wasserstein distance.

## 3.2 Selecting an importance sampling distribution $\nu$

Our next goal is to select an R$\Phi$SD proposal distribution $\nu$ for which the R$\Phi$SD is close to its reference $\Phi$SD even when the importance sample size $M$ is small. Our strategy is to choose $\nu$ so that the second moment of each R$\Phi$SD feature, $w_d(Z, Q_N) := |(Q_N \mathcal{T}_d \Phi)(Z)|^r / \nu(Z)$, is bounded by a power of its mean:

**Definition 3.5** (($C, \gamma$) second moments). *Fix a target distribution $P$. For $Z \sim \nu$, $d \in [D]$, and $N \geq 1$, let $Y_{N,d} := w_d(Z, Q_N)$. If for some $C > 0$ and $\gamma \in [0,2]$ we have $\mathbb{E}[Y_{N,d}^2] \leq C\mathbb{E}[Y_{N,d}]^{2-\gamma}$ for all $d \in [D]$ and $N \geq 1$, then we say $(\Phi, r, \nu)$ yields $(C, \gamma)$ second moments for $P$ and $Q_N$.*

The next proposition, proved in Appendix D, demonstrates the value of this second moment property.

**Proposition 3.6.** *Suppose $(\Phi, r, \nu)$ yields $(C, \gamma)$ second moments for $P$ and $Q_N$. If $M \geq 2C\mathbb{E}[Y_{N,d}]^{-\gamma} \log(D/\delta)/\epsilon^2$ for all $d \in [D]$, then, with probability at least $1 - \delta$,*

$$\mathrm{R}\Phi\mathrm{SD}_{\Phi,r,\nu,M}(Q_N, P) \geq (1 - \epsilon)^{1/r} \, \Phi\mathrm{SD}_{\Phi,r}(Q_N, P).$$

*Under the further assumptions of Proposition 3.1, if the reference $\mathrm{KSD}_k(Q_N, P) \gtrsim N^{-1/2}$,[3] then a sample size $M \gtrsim N^{\gamma r/2} C \|\rho\|_{L^t}^{\gamma r/2} \log(D/\delta)/\epsilon^2$ suffices to have, with probability at least $1 - \delta$,*

$$\|\rho\|_{L^t}^{1/2} \, \mathrm{R}\Phi\mathrm{SD}_{\Phi,r,\nu,M}(Q_N, P) \geq (1 - \epsilon)^{1/r} \, \mathrm{KSD}_k(Q_N, P).$$

Notably, a smaller $r$ leads to substantial gains in the sample complexity $M = \Omega(N^{\gamma r/2})$. For example, if $r = 1$, it suffices to choose $M = \Omega(N^{1/2})$ whenever the weight function $w_d$ is bounded (so that $\gamma = 1$); in contrast, existing analyses of random Fourier features [15, 22, 25, 26, 30] require $M = \Omega(N)$ to achieve the same error rates. We will ultimately show how to select $\nu$ so that $\gamma$ is arbitrarily close to 0. First, we provide simple conditions and a choice for $\nu$ which guarantee $(C, 1)$ second moments.

**Proposition 3.7.** *Assume that $P \in \mathcal{P}$, Assumptions A and B hold with $s = 0$, and there exists a constant $\mathcal{C}' > 0$ such that for all $N \geq 1$, $Q_N([1 + \|\cdot\|]A_N) \leq \mathcal{C}'$. If $\nu(z) \propto Q_N([1 + \|\cdot\|]\Phi(\cdot, z))$, then for any $r \geq 1$, $(\Phi, r, \nu)$ yields $(C, 1)$ second moments for $P$ and $Q_N$.*

Proposition 3.7, which is proved in Appendix E, is based on showing that the weight function $w_d(z, Q_N)$ is uniformly bounded. In order to obtain $(C, \gamma)$ moments for $\gamma < 1$, we will choose $\nu$ such that $w_d(z, Q_N)$ decays sufficiently quickly as $\|z\| \to \infty$. We achieve this by choosing an overdispersed $\nu$—that is, we choose $\nu$ with heavy tails compared to $F$. We also require two integrability conditions involving the Fourier transforms of $\Psi$ and $F$.

**Assumption D.** Assumptions A and B hold, $\omega_1^2 \hat{\Psi}^{1/2}(\omega) \in L^1$, and for $t = r/(2 - r)$, $\hat{\Psi}/\hat{F}^2 \in L^t$.

The $L^1$ condition is an easily satisfied technical condition while the $L^t$ condition ensures that the KSD-$\Phi$SD inequality (2) applies to our chosen $\Phi$SD.

**Theorem 3.8.** *Assume that $P \in \mathcal{P}$, Assumptions A to D hold, and there exists $\mathcal{C} > 0$ such that,*

$$Q_N([1 + \|\cdot\| + \|\cdot - m_N\|^s]A_N / F(\cdot - m_N)) \leq \mathcal{C} \quad \text{for all} \quad N \geq 1. \tag{3}$$

*Then there is a constant $b \in [0, 1)$ such that the following holds. For any $\xi \in (0, 1 - b)$, $c > 0$, and $\alpha > 2(1 - \overline{\lambda})$, if $\nu(z) \geq c\,\Psi(z - m_N)^{\xi r}$, then there exists a constant $C_\alpha > 0$ such that $(\Phi, r, \nu)$ yields $(C_\alpha, \gamma_\alpha)$ second moments for $P$ and $Q_N$, where $\gamma_\alpha := \alpha + (2 - \alpha)\xi/(2 - b - \xi)$.*

Theorem 3.8 suggests a strategy for improving the importance sample growth rate $\gamma$ of an R$\Phi$SD: increase the smoothness $\overline{\lambda}$ of $F$ and decrease the over-dispersion parameter $\xi$ of $\nu$.

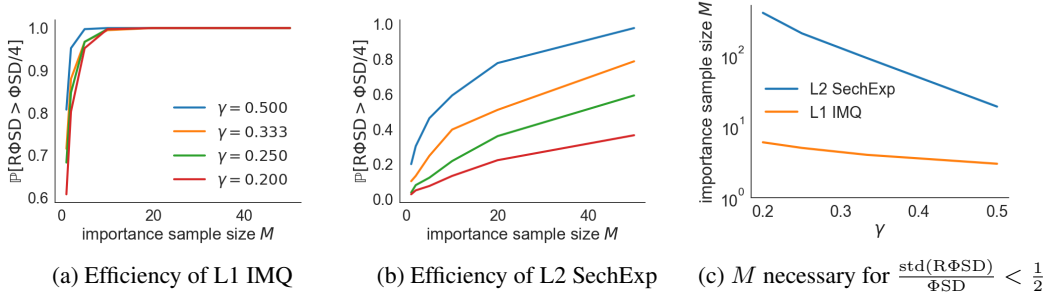

Figure 1: Efficiency of RΦSDs. The L1 IMQ RΦSD displays exceptional efficiency.

### 3.3 Example RΦSDs

In our experiments, we will consider two RΦSDs that determine convergence by Propositions 3.1 and 3.3 and that yield $(C, \gamma)$ second moments for any $\gamma \in (0, 1]$ using Theorem 3.8.

**Example 3.3** ($L^2$ tilted hyperbolic secant RΦSD). Mimicking the construction of the hyperbolic secant kernel in Example 3.2 and following the intuition that $F$ should behave like the square root of $\Psi$, we choose $F = \Psi_{2a}^{\mathrm{sech}}$. As shown in Appendix I, if we choose $r = 2$ and $\nu(z) \propto \Psi_{4a\xi}^{\mathrm{sech}}(z - m_N)$ we can verify all the assumptions necessary for Theorem 3.8 to hold. Moreover, the theorem holds for any $b > 0$ and hence any $\xi \in (0, 1)$ may be chosen. Note that $\nu$ can be sampled from efficiently using the inverse CDF method.

**Example 3.4** ($L^r$ IMQ RΦSD). We can also parallel the construction of the reference IMQ kernel $k(x, y) = \Psi_{c,\beta}^{\mathrm{IMQ}}(x - y)$ from Example 3.1, where $c > 0$ and $\beta \in [-D/2, 0)$. (Recall we have $A = 1$ in Assumption A.) In order to construct a corresponding RΦSD we must choose the constant $\overline{\lambda} \in (1/2, 1)$ that will appear in Assumption C and $\underline{\xi} \in (0, 1/2)$, the minimum $\xi$ we will be able to choose when constructing $\nu$. We show in Appendix J that if we choose $F = \Psi_{c',\beta'}^{\mathrm{IMQ}}$, then Assumptions A to D hold when $c' = \overline{\lambda}c/2$, $\beta' \in [-D/(2\underline{\xi}), -\beta/(2\underline{\xi}) - D/(2\underline{\xi}))$, $r = -D/(2\beta'\underline{\xi})$, $\xi \in (\underline{\xi}, 1)$, and $\nu(z) \propto \Psi_{c',\beta'}^{\mathrm{IMQ}}(z - m_N)^{\xi r}$. A particularly simple setting is given by $\beta' = -D/(2\underline{\xi})$, which yields $r = 1$. Note that $\nu$ can be sampled from efficiently since it is a multivariate $t$-distribution.

In the future it would be interesting to construct other RΦSDs. We can recommend the following fairly simple default procedure for choosing an RΦSD based on a reference KSD admitting the form in Assumption A. (1) Choose any $\gamma > 0$, and set $\alpha = \gamma/3$, $\overline{\lambda} = 1 - \alpha/2$, and $\xi = 4\alpha/(2 + \alpha)$. These are the settings we will use in our experiments. It may be possible to initially skip this step and reason about general choices of $\gamma$, $\xi$, and $\overline{\lambda}$. (2) Pick any $F$ that satisfies $\hat{F}/\hat{\Psi}^{\lambda/2} \in L^2$ for some $\lambda \in (1/2, \overline{\lambda})$ (that is, Assumption C holds) while also satisfying $\hat{\Psi}/\hat{F}^2 \in L^t$ for some $t \in [1, \infty]$. The selection of $t$ induces a choice of $r$ via Assumption D. A simple choice for $F$ is $\mathscr{F}^{-1}\hat{\Psi}^\lambda$. (3) Check if Assumption B holds (it usually does if $F$ decays no faster than a Gaussian); if it does not, a slightly different choice of $F$ should be made. (4) Choose $\nu(z) \propto \Psi(z - m_N)^{\xi r}$.

## 4 Goodness-of-fit testing with RΦSDs

We now detail additional properties of RΦSDs relevant to testing goodness of fit. In goodness-of-fit testing, the sample points $(X_n)_{n=1}^N$ underlying $Q_N$ are assumed to be drawn i.i.d. from a distribution $\mu$, and we wish to use the test statistic $F_{r,N} := \mathrm{R}\Phi\mathrm{SD}_{\Phi,r,\nu,M}^2(Q_N, P)$ to determine whether the null hypothesis $H_0 : P = \mu$ or alternative hypothesis $H_1 : P \neq \mu$ holds. For this end, we will restrict our focus to real analytic $\Phi$ and strictly positive analytic $\nu$, as by Chwialkowski et al. [6, Prop. 2 and Lemmas 1-3], with probability 1, $P = \mu \Leftrightarrow \mathrm{R}\Phi\mathrm{SD}_{\Phi,r,\nu,M}(\mu, P) = 0$ when these properties hold. Thus, analytic RΦSDs do not suffer from the shortcoming of RFFs—which are unable to distinguish between infinitely many distributions with high probability [6].

It remains to estimate the distribution of the test statistic $F_{r,N}$ under the null hypothesis and to verify that the power of a test based on this distribution approaches 1 as $N \to \infty$. To state our result, we assume that $M$ is fixed. Let $\xi_{r,N,dm}(x) := (\mathcal{T}_d\Phi)(x, Z_{N,m})/(M\nu(Z_{N,m}))^{1/r}$ for $r \in [1, 2]$, where

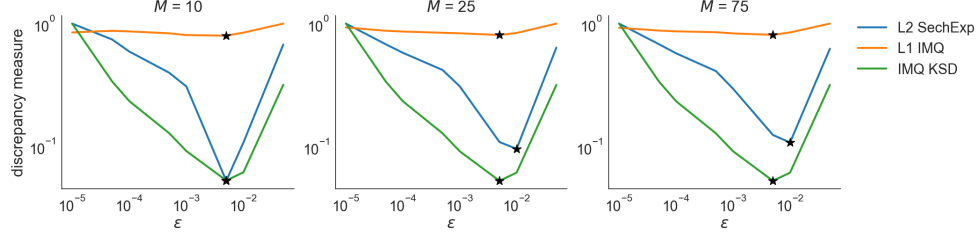

(a) Step size selection using RΦSDs and quadratic-time KSD baseline. With $M \geq 10$, each quality measure selects a step size of $\varepsilon = .01$ or $.005$.

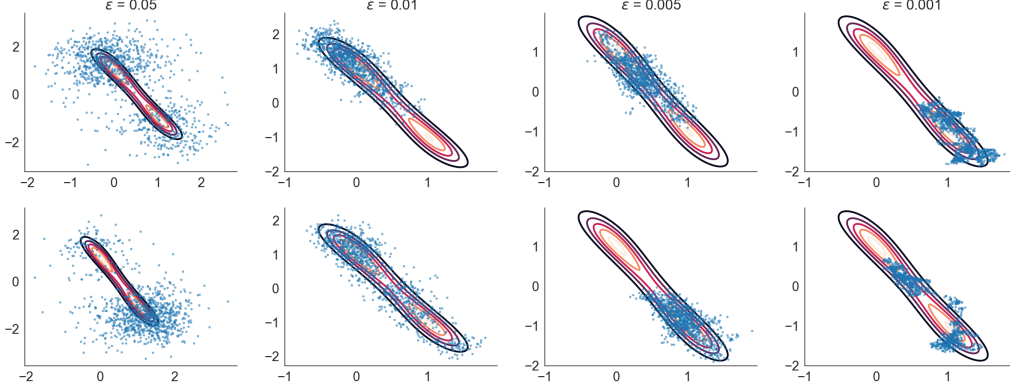

(b) SGLD sample points with equidensity contours of $p$ overlaid. The samples produced by SGLD with $\varepsilon = .01$ or $.005$ are noticeably better than those produced using smaller or large step sizes.

Figure 2: Hyperparameter selection for stochastic gradient Langevin dynamics (SGLD)

$Z_{N,m} \overset{\text{indep}}{\sim} \nu_N$, so that $\xi_{r,N}(x) \in \mathbb{R}^{DM}$. The following result, proved in Appendix K, provides the basis for our testing guarantees.

**Proposition 4.1** (Asymptotic distribution of RΦSD). *Assume $\Sigma_{r,N} := \mathrm{Cov}_P(\xi_{r,N})$ is finite for all $N$ and $\Sigma_r := \lim_{N \to \infty} \Sigma_{r,N}$ exists. Let $\zeta \sim \mathcal{N}(0, \Sigma_r)$. Then as $N \to \infty$: (1) under $H_0 : P = \mu$, $NF_{r,N} \overset{\mathcal{D}}{\Longrightarrow} \sum_{d=1}^{D}(\sum_{m=1}^{M} |\zeta_{dm}|^r)^{2/r}$ and (2) under $H_1 : P \neq \mu$, $NF_{r,N} \overset{P}{\to} \infty$.*

*Remark* 4.2. The condition $\Sigma_r := \lim_{N \to \infty} \Sigma_{r,N}$ holds if $\nu_N = \nu_0(\cdot - m_N)$ for a distribution $\nu_0$.

Our second asympotic result provides a roadmap for using RΦSDs for hypothesis testing and is similar in spirit to Theorem 3 from Jitkrittum et al. [16]. In particular, it furnishes an asymptotic null distribution and establishes asymptotically full power.

**Theorem 4.3** (Goodness of fit testing with RΦSD). *Let $\hat{\mu} := N^{-1} \sum_{n=1}^{N} \xi_{r,N}(X'_n)$ and $\hat{\Sigma} := N^{-1} \sum_{n=1}^{N} \xi_{r,N}(X'_n)\xi_{r,N}(X'_n)^\top - \hat{\mu}\hat{\mu}^\top$ with either $X'_n = X_n$ or $X'_n \overset{i.i.d.}{\sim} P$. Suppose for the test $NF_{r,N}$, the test threshold $\tau_\alpha$ is set to the $(1-\alpha)$-quantile of the distribution of $\sum_{d=1}^{D}(\sum_{m=1}^{M} |\zeta_{dm}|^r)^{2/r}$, where $\zeta \sim \mathcal{N}(0, \hat{\Sigma})$. Then, under $H_0 : P = \mu$, asymptotically the false positive rate is $\alpha$. Under $H_1 : P \neq \mu$, the test power $\mathbb{P}_{H_1}(NF_{r,N} > \tau_\alpha) \to 1$ as $N \to \infty$.*

## 5 Experiments

We now investigate the importance-sample and computational efficiency of our proposed RΦSDs and evaluate their benefits in MCMC hyperparameter selection and goodness-of-fit testing.[4] In our experiments, we considered the RΦSDs described in Examples 3.3 and 3.4: the tilted sech kernel using $r = 2$ and $A(x) = \prod_{d=1}^{D} e^{a'\sqrt{1+x_d^2}}$ (L2 SechExp) and the inverse multiquadric kernel using $r = 1$ (L1 IMQ). We selected kernel parameters as follows. First we chose a target $\gamma$ and then selected $\overline{\lambda}$, $\alpha$, and $\xi$ in accordance with the theory of Section 3 so that $(\Phi, r, \nu)$ yielded $(C_\gamma, \gamma)$

second moments. In particular, we chose $\alpha = \gamma/3$, $\overline{\lambda} = 1 - \alpha/2$, and $\xi = 4\alpha/(2 + \alpha)$. Except for the importance sample efficiency experiments, where we varied $\gamma$ explicitly, all experiments used $\gamma = 1/4$. Let $\widehat{\mathrm{med}}_u$ denote the estimated median of the distance between data points under the $u$-norm, where the estimate is based on using a small subsample of the full dataset. For L2 SechExp, we took $a^{-1} = \sqrt{2\pi}\,\widehat{\mathrm{med}}_1$, except in the sample quality experiments where we set $a^{-1} = \sqrt{2\pi}$. Finding hyperparameter settings for the L1 IMQ that were stable across dimension and appropriately controlled the size for goodness-of-fit testing required some care. However, we can offer some basic guidelines. We recommend choosing $\xi = D/(D + df)$, which ensures $\nu$ has $df$ degrees of freedom. We specifically suggest using $df \in [0.5, 3]$ so that $\nu$ is heavy-tailed no matter the dimension. For most experiments we took $\beta = -1/2$, $c = 4\,\widehat{\mathrm{med}}_2$, and $df = 0.5$. The exceptions were in the sample quality experiments, where we set $c = 1$, and the restricted Boltzmann machine testing experiment, where we set $c = 10\,\widehat{\mathrm{med}}_2$ and $df = 2.5$. For goodness-of-fit testing, we expect appropriate choices for $c$ and $df$ will depend on the properties of the null distribution.

**Importance sample efficiency**   To validate the importance sample efficiency theory from Sections 3.2 and 3.3, we calculated $\mathbb{P}[\mathrm{R\Phi SD} > \Phi\mathrm{SD}/4]$ as the importance sample size $M$ was increased. We considered choices of the parameters for L2 SechExp and L1 IMQ that produced $(C_\gamma, \gamma)$ second moments for varying choices of $\gamma$. The results, shown in Figs. 1a and 1b, indicate greater sample efficiency for L1 IMQ than L2 SechExp. L1 IMQ is also more robust to the choice of $\gamma$. Fig. 1c, which plots the values of $M$ necessary to achieve $\mathrm{stdev}(\mathrm{R\Phi SD})/\Phi\mathrm{SD} < 1/2$, corroborates the greater sample efficiency of L1 IMQ.

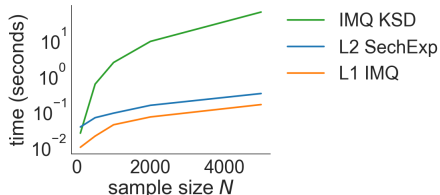

Figure 3: Speed of IMQ KSD vs. R$\Phi$SDs with $M = 10$ importance sample points (dimension $D = 10$). Even for moderate sample sizes $N$, the R$\Phi$SDs are orders of magnitude faster than the KSD.

**Computational complexity**   We compared the computational complexity of the R$\Phi$SDs (with $M = 10$) to that of the IMQ KSD. We generated datasets of dimension $D = 10$ with the sample size $N$ ranging from 500 to 5000. As seen in Fig. 3, even for moderate dataset sizes, the R$\Phi$SDs are computed orders of magnitude faster than the KSD. Other R$\Phi$SDs like FSSD and RFF obtain similar speed-ups; however, we will see the power benefits of the L1 IMQ and L2 SechExp R$\Phi$SDs below.

**Approximate MCMC hyperparameter selection**   We follow the stochastic gradient Langevin dynamics [SGLD, 28] hyperparameter selection setup from Gorham and Mackey [10, Section 5.3]. SGLD with constant step size $\varepsilon$ is a biased MCMC algorithm that approximates the overdamped Langevin diffusion. No Metropolis-Hastings correction is used, and an unbiased estimate of the score function from a data subsample is calculated at each iteration. There is a bias-variance tradeoff in the choice of step size parameter: the stationary distribution of SGLD deviates more from its target as $\varepsilon$ grows, but as $\varepsilon$ gets smaller the mixing speed of SGLD decreases. Hence, an appropriate choice of $\varepsilon$ is critical for accurate posterior inference. We target the bimodal Gaussian mixture model (GMM) posterior of Welling and Teh [28] and compare the step size selection made by the two R$\Phi$SDs to that of IMQ KSD [11] when $N = 1000$. Fig. 2a shows that L1 IMQ and L2 SechExp agree with IMQ KSD (selecting $\varepsilon = .005$) even with just $M = 10$ importance samples. L1 IMQ continues to select $\varepsilon = .005$ while L2 SechExp settles on $\varepsilon = .01$, although the value for $\varepsilon = .005$ is only slightly larger. Fig. 2b compares the choices of $\varepsilon = .005$ and $.01$ to smaller and larger values of $\varepsilon$. The values of $M$ considered all represent substantial reductions in computation as the R$\Phi$SD replaces the $DN(N + 1)/2$ KSD kernel evaluations of the form $((\mathcal{T}_d \otimes \mathcal{T}_d)k)(x_n, x_{n'})$ with only $DNM$ feature function evaluations of the form $(\mathcal{T}_d\Phi)(x_n, z_m)$.

**Goodness-of-fit testing**   Finally, we investigated the performance of R$\Phi$SDs for goodness-of-fit testing. In our first two experiments we used a standard multivariate Gaussian $p(x) = \mathcal{N}(x \mid 0, I)$ as the null distribution while varying the dimension of the data. We explored the power of R$\Phi$SD-based tests compared to FSSD [16] (using the default settings in their code), RFF [22] (Gaussian and Cauchy kernels with bandwidth = $\widehat{\mathrm{med}}_2$), and KSD-based tests [7, 11, 19] (Gaussian kernel with bandwidth

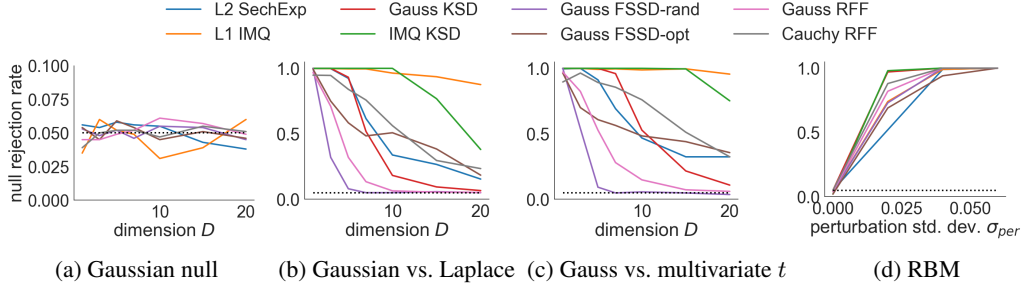

Figure 4: Quadratic-time KSD and linear-time RΦSD, FSSD, and RFF goodness-of-fit tests with $M = 10$ importance sample points (see Section 5 for more details). All experiments used $N = 1000$ except the multivariate $t$, which used $N = 2000$. **(a)** Size of tests for Gaussian null. **(b, c, d)** Power of tests. Both RΦSDs offer competitive performance.

$= \widehat{\text{med}}_2$ and IMQ kernel $\Psi^{\text{IMQ}}_{1,-1/2}$). We did not consider other linear-time KSD approximations due to relatively poor empirical performance [16]. There are two types of FSSD tests: FSSD-rand uses random sample locations and fixed hyperparameters while FSSD-opt uses a small subset of the data to optimize sample locations and hyperparameters for a power criterion. All linear-time tests used $M = 10$ features. The target level was $\alpha = 0.05$. For each dimension $D$ and RΦSD-based test, we chose the nominal test level by generating 200 p-values from the Gaussian asymptotic null, then setting the nominal level to the minimum of $\alpha$ and the 5th percentile of the generated p-values. All other tests had nominal level $\alpha$. We verified the size of the FSSD, RFF, and RΦSD-based tests by generating 1000 p-values for each experimental setting in the Gaussian case (see Fig. 4a). Our first experiment replicated the Gaussian vs. Laplace experiment of Jitkrittum et al. [16] where, under the alternative hypothesis, $q(x) = \prod_{d=1}^{D} \text{Lap}(x_d|0, 1/\sqrt{2})$, a product of Laplace distributions with variance 1 (see Fig. 4b). Our second experiment, inspired by the Gaussian vs. multivariate $t$ experiment of Chwialkowski et al. [7], tested the alternative in which $q(x) = \mathcal{T}(x|0, 5)$, a standard multivariate $t$-distribution with 5 degrees of freedom (see Fig. 4c). Our final experiment replicated the restricted Boltzmann machine (RBM) experiment of Jitkrittum et al. [16] in which each entry of the matrix used to define the RBM was perturbed by independent additive Gaussian noise (see Fig. 4d). The amount of noise was varied from $\sigma_{per} = 0$ (that is, the null held) up to $\sigma_{per} = 0.06$. The L1 IMQ test performed well across all dimensions and experiments, with power of at least 0.93 in almost all experiments. The only exceptions were the Laplace experiment with $D = 20$ (power $\approx 0.88$) and the RBM experiment with $\sigma_{per} = 0.02$ (power $\approx 0.74$). The L2 SechExp test performed comparably to or better than the FSSD and RFF tests. Despite theoretical issues, the Cauchy RFF was competitive with the other linear-time methods—except for the superior L1 IMQ. Given its superior power control and computational efficiency, we recommend the L1 IMQ over the L2 SechExp.

## 6 Discussion and related work

In this paper, we have introduced feature Stein discrepancies, a family of computable Stein discrepancies that can be cheaply approximated using importance sampling. Our stochastic approximations, random feature Stein discrepancies (RΦSDs), combine the computational benefits of linear-time discrepancy measures with the convergence-determining properties of quadratic-time Stein discrepancies. We validated the benefits of RΦSDs on two applications where kernel Stein discrepancies have shown excellent performance: measuring sample quality and goodness-of-fit testing. Empirically, the L1 IMQ RΦSD performed particularly well: it outperformed existing linear-time KSD approximations in high dimensions and performed as well or better than the state-of-the-art quadratic-time KSDs.

RΦSDs could also be used as drop-in replacements for KSDs in applications to Monte Carlo variance reduction with control functionals [21], probabilistic inference using Stein variational gradient descent [18], and kernel quadrature [2, 3]. Moreover, the underlying principle used to generalize the KSD could also be used to develop fast alternatives to maximum mean discrepancies in two-sample testing applications [6, 13]. Finally, while we focused on the Langevin Stein operator, our development is compatible with any Stein operator, including diffusion Stein operators [12].

**Acknowledgments**

Part of this work was done while JHH was a research intern at MSR New England.

## Footnotes

[2]We say $P$ satisfies *distant dissipativity* [8, 12] if $\kappa_0 := \liminf_{r \to \infty} \kappa(r) > 0$ for $\kappa(r) = \inf\{-2\langle \nabla \log p(x) - \nabla \log p(y), x - y \rangle / \|x - y\|_2^2 : \|x - y\|_2 = r\}$.

[3]Note that $\mathrm{KSD}_k(Q_N, P) = \Omega_P(N^{-1/2})$ whenever the sample points $x_1, \ldots, x_N$ are drawn i.i.d. from a distribution $\mu$, since the scaled V-statistic $N\,\mathrm{KSD}_k^2(Q_N, P)$ diverges when $\nu \neq P$ and converges in distribution to a non-zero limit when $\nu = P$ [23, Thm. 32]. Moreover, working in a hypothesis testing framework of shrinking alternatives, Gretton et al. [13, Thm. 13] showed that $\mathrm{KSD}_k(Q_N, P) = \Theta(N^{-1/2})$ was the smallest local departure distinguishable by an asymptotic KSD test.

[4]See https://bitbucket.org/jhhuggins/random-feature-stein-discrepancies for our code.

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
