[Supplementary Material · fast_kernels_appendix.pdf]

# Appendix for "Random Feature Stein Discrepancies"

## A  Proof of Proposition 3.1: KSD-$\Phi$SD inequality

We apply the generalized Hölder's inequality and the Babenko-Beckner inequality in turn to find

$$
\begin{aligned}
\mathrm{KSD}_k^2(Q_N, P) &= \sum_{d=1}^D \int |\mathscr{F}(Q_N(\mathcal{T}_d\Phi))(\omega)|^2 \rho(\omega)\, d\omega \le \|\rho\|_{L^t} \sum_{d=1}^D \|\mathscr{F}(Q_N(\mathcal{T}_d\Phi))\|_{L^s}^2 \\
&\le c_{r,d}^2 \|\rho\|_{L^t} \sum_{d=1}^D \|Q_N(\mathcal{T}_d\Phi)\|_{L^r}^2 = c_{r,d}^2 \|\rho\|_{L^t} \, \Phi\mathrm{SD}_{\Phi,r}^2(Q_N, P),
\end{aligned}
$$

where $t = \frac{r}{2-r}$ and $c_{r,d} := (r^{1/r}/s^{1/s})^{d/2} \le 1$ for $s = r/(r-1)$.

## B  Proof of Theorem 3.2: Tilted KSDs detect non-convergence

For any vector-valued function $f$, let $M_1(f) = \sup_{x,y:\|x-y\|_2=1} \|f(x) - f(y)\|_2$. The result will follow from the following theorem which provides an upper bound on the bounded Lipschitz metric $d_{BL_{\|\cdot\|_2}}(\mu, P)$ in terms of the KSD and properties of $A$ and $\Psi$. Let $b := \nabla \log p$.

**Theorem B.1** (Tilted KSD lower bound). *Suppose $P \in \mathcal{P}$ and $k(x, y) = A(x)\Psi(x - y)A(y)$ for $\Psi \in C^2$ and $A \in C^1$ with $A > 0$ and $\nabla \log A$ bounded and Lipschitz. Then there exists a constant $\mathcal{M}_P$ such that, for all $\epsilon > 0$ and all probability measures $\mu$,*

$$
d_{BL_{\|\cdot\|_2}}(\mu, P) \le \epsilon + C \, \mathrm{KSD}_k(\mu, P),
$$

*where*

$$
C := (2\pi)^{-d/4} \|1/A\|_{L^2} \mathcal{M}_P H \big( \mathbb{E}[\|G\|_2 B(G)](1 + M_1(\log A) + \mathcal{M}_P M_1(b + \nabla \log A))\epsilon^{-1} \big)^{1/2},
$$

$H(t) := \sup_{\omega \in \mathbb{R}^d} e^{-\|\omega\|_2^2/(2t^2)}/\hat{\Psi}(\omega)$, *$G$ is a standard Gaussian vector, and* $B(y) := \sup_{x \in \mathbb{R}^d, u \in [0,1]} A(x)/A(x + uy)$.

**Remarks**    By bounding $H$ and optimizing over $\epsilon$, one can derive rates of convergence in $d_{BL_{\|\cdot\|_2}}$. Thm. 5 and Sec. 4.2 of Gorham et al. [12] provide an explicit value for the *Stein factor* $\mathcal{M}_P$.

Let $A_\mu(x) = A(x - \mathbb{E}_{X\sim\mu}[X])$. Since $\|1/A\|_{L^2} = \|1/A_\mu\|_{L^2}$, $M_1(\log A_\mu) \le M_1(\log A)$, $M_1(\nabla \log A_\mu) \le M_1(\nabla \log A)$, and $\sup_{x \in \mathbb{R}^d, u \in [0,1]} A_\mu(x)/A_\mu(x + uy) = B(y)$, the exact conclusion of Theorem B.1 also holds when $k(x, y) = A_\mu(x)\Psi(x - y)A_\mu(y)$. Moreover, since $\log A$ is Lipschitz, $B(y) \le e^{\|y\|_2}$ so $\mathbb{E}[\|G\|_2 B(G)]$ is finite. Now suppose $\mathrm{KSD}_k(\mu_N, P) \to 0$ for a sequence of probability measures $(\mu_N)_{N\ge1}$. For any $\epsilon > 0$, $\limsup_n d_{BL_{\|\cdot\|_2}}(\mu_N, P) \le \epsilon$, since $H(t)$ is finite for all $t > 0$. Hence, $d_{BL_{\|\cdot\|_2}}(\mu_N, P) \to 0$, and, as $d_{BL_{\|\cdot\|_2}}$ metrizes weak convergence, $\mu_N \Rightarrow P$.

### B.1  Proof of Theorem B.1: Tilted KSD lower bound

Our proof parallels that of [11, Thm. 13]. Fix any $h \in BL_{\|\cdot\|_2}$. Since $A \in C^1$ is positive, Thm. 5 and Sec. 4.2 of Gorham et al. [12] imply that there exists a $g \in C^1$ which solves the Stein equation $\mathcal{T}_P(Ag) = h - \mathbb{E}_P[h(Z)]$ and satisfies $M_0(Ag) \le \mathcal{M}_P$ for $\mathcal{M}_P$ a constant independent of $A, h$, and $g$. Since $1/A \in L^2$, we have $\|g\|_{L^2} \le \mathcal{M}_P \|1/A\|_{L^2}$.

Since $\nabla \log A$ is bounded, $A(x) \le \exp(\gamma\|x\|)$ for some $\gamma$. Moreover, any measure in $\mathcal{P}$ is sub-Gaussian, so $P$ has finite exponential moments. Hence, since $A$ is also positive, we may define the tilted probability measure $P_A$ with density proportional to $Ap$. The identity $\mathcal{T}_P(Ag) = A\mathcal{T}_{P_A}g$ implies that

$$
M_0(A\nabla\mathcal{T}_{P_A}g) = M_0(\nabla\mathcal{T}_P(Ag) - \mathcal{T}_P(Ag)\nabla\log A) \le 1 + M_1(\log A).
$$

Since $b$ and $\nabla \log A$ are Lipschitz, we may apply the following lemma, proved in Appendix B.2 to deduce that there is a function $g_\epsilon \in \mathcal{K}_{k_1}^d$ for $k_1(x, y) := \Psi(x - y)$ such that $|(\mathcal{T}_P(Ag_\epsilon))(x) - (\mathcal{T}_P(Ag))(x)| = A(x)|(\mathcal{T}_{P_A}g_\epsilon)(x) - (\mathcal{T}_{P_A}g)(x)| \le \epsilon$ for all $x$ with norm

$$
\|g_\epsilon\|_{\mathcal{K}_{k_1}^d} \tag{4}
$$

$$
\le (2\pi)^{-d/4} H \big( \mathbb{E}[\|G\|_2 B(G)](1 + M_1(\log A) + \mathcal{M}_P M_1(b + \nabla \log A))\epsilon^{-1} \big)^{1/2} \|1/A\|_{L^2} \mathcal{M}_P.
$$

**Lemma B.2** (Stein approximations with finite RKHS norm). *Consider a function $A : \mathbb{R}^d \to \mathbb{R}$ satisfying $B(y) := \sup_{x \in \mathbb{R}^d, u \in [0,1]} A(x)/A(x+uy)$. Suppose $g : \mathbb{R}^d \to \mathbb{R}^d$ is in $L^2 \cap C^1$. If $P$ has Lipschitz log density, and $k(x,y) = \Psi(x-y)$ for $\Psi \in C^2$ with generalized Fourier transform $\hat{\Psi}$, then for every $\epsilon \in (0,1]$, there is a function $g_\epsilon : \mathbb{R}^d \to \mathbb{R}^d$ such that $|(\mathcal{T}_P g_\epsilon)(x) - (\mathcal{T}_P g)(x)| \le \epsilon/A(x)$ for all $x \in \mathbb{R}^d$ and*

$$\|g_\epsilon\|_{\mathcal{K}_k^d} \le (2\pi)^{-d/4} H\left(\mathbb{E}[\|G\|_2 B(G)](M_0(A\nabla \mathcal{T}_P g) + M_0(Ag)M_1(b))\epsilon^{-1}\right)^{1/2} \|g\|_{L^2},$$

*where $H(t) := \sup_{\omega \in \mathbb{R}^d} e^{-\|\omega\|_2^2/(2t^2)}/\hat{\Psi}(\omega)$ and $G$ is a standard Gaussian vector.*

Since $\|Ag_\epsilon\|_{\mathcal{K}_k^d} = \|g_\epsilon\|_{\mathcal{K}_{k_1}^d}$, the triangle inequality and the definition of the KSD now yield

$$
\begin{aligned}
|\mathbb{E}_\mu[h(X)] - \mathbb{E}_P[h(Z)]| &= |\mathbb{E}_\mu[(\mathcal{T}_P(Ag))(X)]| \\
&\le |\mathbb{E}[(\mathcal{T}_P(Ag))(X) - (\mathcal{T}_P(Ag_\epsilon))(X)]| + |\mathbb{E}_\mu[(\mathcal{T}_P(Ag_\epsilon))(X)]| \\
&\le \epsilon + \|g_\epsilon\|_{\mathcal{K}_{k_1}^d} \; \mathrm{KSD}_k(\mu, P).
\end{aligned}
$$

The advertised conclusion follows by applying the bound (4) and taking the supremum over all $h \in BL_{\|\cdot\|}$.

## B.2 Proof of Lemma B.2: Stein approximations with finite RKHS norm

Assume $M_0(A\nabla \mathcal{T}_P g) + M_0(Ag) < \infty$, as otherwise the claim is vacuous. Our proof parallels that of Gorham and Mackey [11, Lem. 12]. Let $Y$ denote a standard Gaussian vector with density $\rho$. For each $\delta \in (0,1]$, we define $\rho_\delta(x) = \delta^{-d}\rho(x/\delta)$, and for any function $f$ we write $f_\delta(x) \triangleq \mathbb{E}[f(x+\delta Y)]$. Under our assumptions on $h = \mathcal{T}_P g$ and $B$, the mean value theorem and Cauchy-Schwarz imply that for each $x \in \mathbb{R}^d$ there exists $u \in [0,1]$ such that

$$
\begin{aligned}
|h_\delta(x) - h(x)| &= |\mathbb{E}_\rho[h(x+\delta Y) - h(x)]| = |\mathbb{E}_\rho[\langle \delta Y, \nabla h(x+\delta Y u)\rangle]| \\
&\le \delta M_0(A\nabla \mathcal{T}_P g) \, \mathbb{E}_\rho[\|Y\|_2/A(x+\delta Y u)] \le \delta M_0(A\nabla \mathcal{T}_P g) \, \mathbb{E}_\rho[\|Y\|_2 B(Y)]/A(x).
\end{aligned}
$$

Now, for each $x \in \mathbb{R}^d$ and $\delta > 0$,

$$
\begin{aligned}
h_\delta(x) &= \mathbb{E}_\rho[\langle b(x+\delta Y), g(x+\delta Y)\rangle] + \mathbb{E}[\langle \nabla, g(x+\delta Y)\rangle] \quad \text{and} \\
(\mathcal{T}_P g_\delta)(x) &= \mathbb{E}_\rho[\langle b(x), g(x+\delta Y)\rangle] + \mathbb{E}[\langle \nabla, g(x+\delta Y)\rangle],
\end{aligned}
$$

so, by Cauchy-Schwarz, the Lipschitzness of $b$, and our assumptions on $g$ and $B$,

$$
\begin{aligned}
|(\mathcal{T}_P g_\delta)(x) - h_\delta(x)| &= |\mathbb{E}_\rho[\langle b(x) - b(x+\delta Y), g(x+\delta Y)\rangle]| \\
&\le \mathbb{E}_\rho[\|b(x) - b(x+\delta Y)\|_2 \|g(x+\delta Y)\|_2] \\
&\le M_0(Ag)M_1(b)\,\delta\, \mathbb{E}_\rho[\|Y\|_2/A(x+\delta Y)] \le M_0(Ag)M_1(b)\,\delta\, \mathbb{E}_\rho[\|Y\|_2 B(Y)]/A(x).
\end{aligned}
$$

Thus, if we fix $\epsilon > 0$ and define $\tilde{\epsilon} = \epsilon/(\mathbb{E}_\rho[\|Y\|_2 B(Y)](M_0(A\nabla \mathcal{T}_P g) + M_0(Ag)M_1(b)))$, the triangle inequality implies

$$|(\mathcal{T}_P g_{\tilde{\epsilon}})(x) - (\mathcal{T}_P g)(x)| \le |(\mathcal{T}_P g_{\tilde{\epsilon}})(x) - h_{\tilde{\epsilon}}(x)| + |h_{\tilde{\epsilon}}(x) - h(x)| \le \epsilon/A(x).$$

To conclude, we will bound $\|g_\delta\|_{\mathcal{K}_k^d}$. By Wendland [29, Thm. 10.21],

$$
\begin{aligned}
\|g_\delta\|_{\mathcal{K}_k^d}^2 &= (2\pi)^{-d/2} \int_{\mathbb{R}^d} \frac{|\hat{g}_\delta(\omega)|^2}{\hat{\Phi}(\omega)} \, d\omega = (2\pi)^{d/2} \int_{\mathbb{R}^d} \frac{|\hat{g}(\omega)|^2 \hat{\rho}_\delta(\omega)^2}{\hat{\Phi}(\omega)} \, d\omega \\
&\le (2\pi)^{-d/2} \left\{\sup_{\omega \in \mathbb{R}^d} \frac{e^{-\|\omega\|_2^2 \delta^2/2}}{\hat{\Phi}(\omega)}\right\} \int_{\mathbb{R}^d} |\hat{g}(\omega)|^2 \, d\omega,
\end{aligned}
$$

where we have used the Convolution Theorem [29, Thm. 5.16] and the identity $\hat{\rho}_\delta(\omega) = \hat{\rho}(\delta\omega)$. Finally, an application of Plancherel's theorem [14, Thm. 1.1] gives $\|g_\delta\|_{\mathcal{K}_k^d} \le (2\pi)^{-d/4} F(\delta^{-1})^{1/2} \|g\|_{L^2}$.

## C  Proof of Proposition 3.3

We begin by establishing the $\Phi$SD convergence claim. Define the target mean $m_P := \mathbb{E}_{Z \sim P}[Z]$. Since $\log A$ is Lipschitz and $A > 0$, $A_N \leq A e^{m_N}$ and hence $P(A_N) < \infty$ and $\mathbb{E}_P\left[A_N(Z)\|Z\|_2^2\right] < \infty$ for all $N$ by our integrability assumptions on $P$.

Suppose $\mathcal{W}_{A_N}(Q_N, P) \to 0$, and, for any probability measure $\mu$ with $\mu(A_N) < \infty$, define the tilted probability measure $\mu_{A_N}$ via $d\mu_{A_N}(x) = d\mu(x)A_N(x)$. By the definition of $\mathcal{W}_{A_N}$, we have $|Q_N(A_N h) - P(A_N h)| \to 0$ for all $h \in \mathcal{H}$. In particular, since the constant function $h(x) = 1$ is in $\mathcal{H}$, we have $|Q_N(A_N) - P(A_N)| \to 0$. In addition, since the functions $f_N(x) = (x - m_N)/A_N(x)$ are uniformly Lipschitz in $N$, we have $m_N - m_P = Q_N(f_N) - P(f_N) \to 0$ and thus $A_N \to A_P$ for $A_P(x) := A(x - m_P) > 0$. Therefore, $P(A_N) \to P(A_P) > 0$, and, as $x/y$ is a continuous function of $(x, y)$ when $y > 0$, we have

$$Q_{N,A_N}(h) - P_{A_N}(h) = Q_N(A_N h)/Q_N(A_N) - P(A_N h)/P(A_N) \to 0$$

and hence the 1-Wasserstein distance $d_{\mathcal{H}}(Q_{N,A_N}, P_{A_N}) \to 0$.

Now note that, for any $g \in \mathcal{G}_{\Phi/A_N, r}$,

$$\begin{aligned}
Q_N(\mathcal{T} A_N g) = Q_N(A_N \mathcal{T}_{P_{A_N}} g) &= Q_N(A_N) Q_{N,A_N}(\mathcal{T}_{P_{A_N}} g) \\
&= ((Q_N(A_N) - P(A_N)) + P(A_N)) Q_{N,A_N}(\mathcal{T}_{P_{A_N}} g) \\
&\leq (\mathcal{W}_{A_N}(Q_N, P) + P(A_N)) Q_{N,A_N}(\mathcal{T}_{P_{A_N}} g)
\end{aligned}$$

where $\mathcal{T}_{P_{A_N}}$ is the Langevin operator for the tilted measure $P_{A_N}$, defined by

$$(\mathcal{T}_{P_{A_N}} g)(x) = \sum_{d=1}^{D} (p(x)A_N(x))^{-1} \partial_{x_d}(p(x)A_N(x)g_d(x)).$$

Taking a supremum over $g \in \mathcal{G}_{\Phi/A_N, r}$, we find

$$\Phi SD_{\Phi, r}(Q_N, P) \leq (\mathcal{W}_{A_N}(Q_N, P) + P(A_N)) \, \Phi SD_{\Phi/A_N, r}(Q_{N,A_N}, P_{A_N}).$$

Furthermore, since $\Phi(x, z)/A_N(x) = F(x - z)$, Hölder's inequality implies

$$\begin{aligned}
\sup_{x \in \mathbb{R}^D} \|g(x)\|_\infty &\leq \|F\|_{L^r}, \\
\sup_{x \in \mathbb{R}^D, d \in [D]} \|\partial_{x_d} g(x)\|_\infty &\leq \|\partial_{x_d} F\|_{L^r}, \quad \text{and} \\
\sup_{x \in \mathbb{R}^D, d, d' \in [D]} \|\partial_{x_d} \partial_{x_{d'}} g(x)\|_\infty &\leq \|\partial_{x_d} \partial_{x_{d'}} F\|_{L^r}
\end{aligned}$$

for each $g \in \mathcal{G}_{\Phi/A_N, r}$. Since $\nabla \log p$ and $\nabla \log A_N$ are Lipschitz and $\mathbb{E}_P\left[A_N(Z)\|Z\|_2^2\right] < \infty$, we may therefore apply [11, Lem. 18] to discover that $\Phi SD_{\Phi/A_N, r}(Q_{N,A_N}, P_{A_N}) \to 0$ and hence $\Phi SD_{\Phi, r}(Q_N, P) \to 0$ whenever the 1-Wasserstein distance $d_{\mathcal{H}}(Q_{N,A_N}, P_{A_N}) \to 0$.

To see that $\mathrm{R}\Phi SD^2_{\Phi, r, \nu_N, M_N}(Q_N, P) \xrightarrow{P} 0$ whenever $\Phi SD^2_{\Phi, r}(Q_N, P) \to 0$, first note that since $r \in [1, 2]$, we may apply Jensen's inequality to obtain

$$\begin{aligned}
\mathbb{E}[\mathrm{R}\Phi SD^2_{\Phi, r, \nu_N, M_N}(Q_N, P)] &= \mathbb{E}[\sum_{d=1}^{D} (\frac{1}{M} \sum_{m=1}^{M} \nu_N(Z_m)^{-1} |Q_N(\mathcal{T}_d \Phi)(Z_m)|^r)^{2/r}] \\
&\leq \sum_{d=1}^{D} (\mathbb{E}[\frac{1}{M} \sum_{m=1}^{M} \nu_N(Z_m)^{-1} |Q_N(\mathcal{T}_d \Phi)(Z_m)|^r])^{2/r} \\
&= \Phi SD^2_{\Phi, r}(Q_N, P).
\end{aligned}$$

Hence, by Markov's inequality, for any $\epsilon > 0$,

$$\mathbb{P}[\mathrm{R}\Phi SD^2_{\Phi, r, \nu_N, M_N}(Q_N, P) > \epsilon] \leq \mathbb{E}[\mathrm{R}\Phi SD^2_{\Phi, r, \nu_N, M_N}(Q_N, P)]/\epsilon \leq \Phi SD^2_{\Phi, r}(Q_N, P)/\epsilon \to 0,$$

yielding the result.

# D   Proof of Proposition 3.6

To achieve the first conclusion, for each $d \in [D]$, apply Corollary M.2 with $\delta/D$ in place of $\delta$ to the random variable

$$\frac{1}{M} \sum_{m=1}^{M} w_d(Z_m, Q_N).$$

The first claim follows by plugging in the high probability lower bounds from Corollary M.2 into $\mathrm{R\Phi SD}^2_{\Phi,r,\nu,M}(Q_N, P)$ and using the union bound.

The equality $\mathbb{E}[Y_d] = \Phi\mathrm{SD}^r_{\Phi,r}(Q_N, P)$, the KSD-$\Phi$SD inequality of Proposition 3.1 ($\Phi\mathrm{SD}^r_{\Phi,r}(Q_N, P) \geq \mathrm{KSD}^r_k(Q_N, P)\|\rho\|_{L^t}^{-r/2}$), and the assumption $\mathrm{KSD}_k(Q_N, P) \gtrsim N^{-1/2}$ imply that $\mathbb{E}[Y_d] \gtrsim N^{-r/2}\|\rho\|_{L^t}^{-r/2}$. Plugging this estimate into the initial importance sample size requirement and applying the KSD-$\Phi$SD inequality once more yield the second claim.

# E   Proof of Proposition 3.7

It turns out that we obtain $(C, 1)$ moments whenever the weight functions $w_d(z, Q_N)$ are bounded. Let $\mathcal{Q}(\Phi, \nu, C') := \{Q_N \mid \sup_{z,d} w_d(z, Q_N) < C'\}$.

**Proposition E.1.** *For any $C > 0$, $(\Phi, r, \nu)$ yields $(C, 1)$ second moments for $P$ and $\mathcal{Q}(\Phi, \nu, C')$.*

**Proof**   It follows from the definition of $\mathcal{Q}(\Phi, \nu, C)$ that

$$\sup_{Q_N \in \mathcal{Q}(\Phi,\nu,C)} \sup_{d,z} |(Q_N \mathcal{T}_d \Phi)(z)|^r / \nu(z) \leq C.$$

Hence for any $Q_N \in \mathcal{Q}(\Phi, \nu, C)$ and $d \in [D]$, $Y_d \leq C$ a.s. and thus

$$\mathbb{E}[Y_d^2] \leq C'\mathbb{E}[Y_d].$$

$\square$

Thus, to prove Proposition 3.7 it suffices to have uniform bound for $w_d(z, Q_N)$ for all $Q_N \in \mathcal{Q}(C')$. Let $\sigma(x) := 1 + \|x\|$ and fix some $Q \in \mathcal{Q}(C')$. Then $\nu(z) = Q_N(\sigma\Phi(\cdot, z))/C(Q_N)$, where $C(Q_N) := \|F\|_{L^1}\mathcal{Q}(\sigma A(\cdot - m_N)) \leq \|F\|_{L^1}\mathcal{C}'$. Moreover, for $c, c' > 0$ not depending on $Q_N$,

$$\begin{aligned}
|(Q_N \mathcal{T}_d \Phi)(z)|^r &\leq Q_N(|\partial_d \log p + \partial_d \log A(\cdot - m_N) + \partial_d \log F(\cdot - z)|\Phi(\cdot, z))^r \\
&\leq cQ_N(|1 + \|\cdot\| + \|\cdot - m_N\|^a|\Phi(\cdot, z))^r \\
&\leq c'(\mathcal{C}')^{r-1}Q_N(\sigma\Phi(\cdot, z)).
\end{aligned}$$

Thus,

$$w_d(z, Q_N) = \frac{|(Q_N \mathcal{T}_d \Phi)(z)|^r}{\nu(z)} \leq \frac{C(\mathcal{Q})c'(\mathcal{C}')^{r-1}Q_N(\sigma\Phi(\cdot, z))}{Q_N(\sigma\Phi(\cdot, z))} \leq c'(\mathcal{C}')^r\|F\|_{L^1}.$$

# F   Technical Lemmas

**Lemma F.1.** *If $P \in \mathcal{P}$, Assumptions A to D hold, and (3) holds, then for any $\lambda \in (1/2, \overline{\lambda})$,*

$$|(Q_N \mathcal{T}_d \Phi)(z)| \leq C_{\lambda,\mathcal{C}} \, \mathrm{KSD}_{k_d}^{2\lambda-1}.$$

**Proof**   Let $\varsigma_d(\omega) := (1 + \omega_d)^{-1}Q_N(\mathcal{T}_d A(\cdot - m_N)e^{-i\omega\cdots})$. Applying Proposition H.1 with $\mathcal{D} = Q_N \mathcal{T}_d A(\cdot - m_N)$, $h = F$, $\varrho(\omega) = 1 + \omega_d$, and $t = 1/2$ yields

$$|(Q_N \mathcal{T}_d \Phi)(z)| \leq \|F\|_{\Psi(\lambda)} \left(\|\varsigma_d\|_{L^\infty}\|(1 + \partial_d)\Psi^{(1/4)}\|_{L^2}\right)^{2-2\lambda} \|Q_N \mathcal{T}_d \Phi\|_{\Psi}^{2\lambda-1}$$

The finiteness of $\|F\|_{\Psi(\lambda)}$ follows from Assumption C. Using $P \in \mathcal{P}$, Assumption A, and (3) we have

$$\begin{aligned}
\varsigma_d(\omega) &= (1 + \omega_d)^{-1}Q_N([\partial_d \log p + \partial_d \log A(\cdot - m_N) - i\omega_d]A(\cdot - m_N)e^{-i\omega\cdots}) \\
&\leq CQ_N([1 + \|\cdot\|]A(\cdot - m_N) \\
&\leq C\mathcal{C}',
\end{aligned}$$

so $\|\varsigma_d\|_{L^\infty}$ is finite. The finiteness of $\|(1 + \partial_d)\Psi^{(1/4)}\|_{L^2}$ follows from the Plancherel theorem and Assumption D. The result now follows upon noting that $\|Q_N \mathcal{T}_d \Phi\|_\Psi = \mathrm{KSD}_{k_d}$. $\qquad\square$

**Lemma F.2.** *If $P \in \mathcal{P}$, Assumptions A and B hold, and (3) holds, then for some $b \in [0, 1), C_b > 0$,*

$$|Q_N \mathcal{T}_d \Phi(z)| \leq C_b F(z - m_N)^{1-b}.$$

*Moreover, $b = 0$ if $s = 0$.*

**Proof**  We have (with $C$ a constant changing line to line)

$$
\begin{aligned}
|Q_N \mathcal{T}_d \Phi(z)| &\leq Q_N |\mathcal{T}_d \Phi(\cdot, z)| \\
&= Q_N(|\partial_d \log p + \partial_d \log A(\cdot - m_N) + \partial_d \log F(\cdot - z)| A(\cdot - m_N) F(\cdot - z)) \\
&\leq C Q_N(1 + \|\cdot\| + \|\cdot - z\|^s | A(\cdot - m_N) F(\cdot - m_N)^{-1}) F(z - m_N) \\
&\leq C Q_N(1 + \|\cdot\| + \|\cdot - m_N\|^s + \|z - m_N\|^s | A(\cdot - m_N) F(\cdot - m_N)^{-1}) F(z - m_N) \\
&\leq C\mathcal{C}(1 + \|z - m_N\|^s) F(z - m_N).
\end{aligned}
$$

By assumption $(1 + \|z\|^s) F(z) \to 0$ as $\|z\| \to \infty$, so for some $C_b > 0$ and $b \in [0, 1)$, $(1 + \|z - m_N\|^s) \leq C_b F(z)^{-b}$. $\qquad\square$

## G  Proof of Theorem 3.8: $(C, \gamma)$ second moment bounds for $\mathrm{R}\Phi\mathrm{SD}$

Take $Q_N \in \mathcal{Q}(\mathcal{C})$ fixed and let $w_d(z) := w_d(z, Q_N)$. For a set $S$ let $\nu_S(S') := \int_{S \cap S'} \nu(\mathrm{d}z)$. Let $K := \{x \in \mathbb{R}^D \mid \|x - m_N\| \leq R\}$. Recall that $Z \sim \nu$ and $Y_d = w_d(Z)$. We have

$$
\begin{aligned}
\mathbb{E}[Y_d^2] &= \mathbb{E}[w_d(Z)^2] = \mathbb{E}[w_d(Z)^2 \mathbb{1}(Z \in K)] + \mathbb{E}[w_d(Z)^2 \mathbb{1}(Z \notin K)] \\
&\leq \|w_d\|_{L^1(\nu)} \|w_d \mathbb{1}(\cdot \in K)\|_{L^\infty(\nu)} + \|\mathbb{1}(\cdot \notin K)\|_{L^1(\nu)} \|w_d^2 \mathbb{1}(\cdot \notin K)\|_{L^\infty(\nu)} \\
&= \|Q_N \mathcal{T}_d \Phi\|_{L^r}^r \sup_{z \in K} w_d(z) + \nu(K^\complement) \sup_{z \in K^\complement} w_d(z)^2 \\
&= \mathbb{E}[Y_d] \sup_{z \in K} w_d(z) + \nu(K^\complement) \sup_{z \in K^\complement} w_d(z)^2
\end{aligned}
$$

Without loss of generality we can take $\nu(z) = \Psi(z - m_N)^{\xi r} / \|\Psi^{\xi r}\|_{L^1}$, since a different choice of $\nu$ only affects constant factors. Applying Lemma F.1, Assumption D, and (2), we have

$$
\begin{aligned}
\sup_{z \in K} w_d(z) &\leq C_{\lambda, \mathcal{C}}^r \mathrm{KSD}_{k_d}^{r(2\lambda - 1)} \sup_{z \in K} \nu(z)^{-1} \\
&\leq C_{\lambda, \mathcal{C}}^r \|\Psi^{\xi r}\|_{L^1} \sup_{z \in K} F(z - m_N)^{-\xi r} \mathrm{KSD}_{k_d}^{r(2\lambda - 1)} \\
&\leq C_{\lambda, \mathcal{C}}^r \underline{c}^{-\xi r} \|\Psi^{\xi r}\|_{L^1} \|\hat\Psi / \hat{F}^2\|_{L^t} f(R)^{-\xi r} \|Q_N \mathcal{T}_d \Phi\|_{L^r}^{r(2\lambda - 1)} \\
&= C_{\lambda, \mathcal{C}}^r \|(\Psi/\underline{c})^{\xi r}\|_{L^1} \|\hat\Psi / \hat{F}^2\|_{L^t} f(R)^{-\xi r} \mathbb{E}[Y_d]^{2\lambda - 1}.
\end{aligned}
$$

Applying Lemma F.2 we have

$$
\begin{aligned}
\sup_{z \in K^\complement} w_d(z)^2 &\leq C_b^2 \sup_{z \in K^\complement} F(z - m_N)^{2(1-b)r} / \nu(z)^2 \\
&= C_b^2 \|\Psi^{\xi r}\|_{L^1}^2 \sup_{z \in K^\complement} F(z - m_N)^{2(1-b-\xi)r} \\
&= C_b^2 \|\Psi^{\xi r}\|_{L^1}^2 f(R)^{2(1-b-\xi)r}.
\end{aligned}
$$

Thus, we have that

$$\mathbb{E}[Y_d^2] \leq C_{\lambda, \mathcal{C}, r, \xi} \mathbb{E}[Y_d]^{2\lambda} f(R)^{-\xi r} + C_{b, \xi r} f(R)^{2(1-b-\xi)r}.$$

As long as $\mathbb{E}[Y_d]^{2\lambda} \leq C_{b, \xi r} f(0)^{2(1-b-\xi/2)r} / C_{\lambda, \mathcal{C}, r, \xi}$, since $f$ is continuous and non-increasing to zero we can choose $R$ such that $f(R)^{2(1-b-\xi)r} = C_{\lambda, \mathcal{C}, r, \xi} \mathbb{E}[Y_d]^{2\lambda} / C_{b, \xi r}$ and the result follows for

$$\mathbb{E}[Y_d]^{2\lambda} \leq C_{b, \xi r} f(0)^{2(1-b-\xi/2)r} / C_{\lambda, \mathcal{C}, r, \xi}.$$

Otherwise, we can guarantee that $\mathbb{E}[Y_d^2] \leq C_\alpha \mathbb{E}[Y_d]^{2-\gamma_\alpha}$ be choosing $C_\alpha$ sufficiently large, since by assumption $\mathbb{E}[Y_d]$ is uniformly bounded over $Q_N \in \mathcal{Q}(\mathcal{C})$.

# H A uniform MMD-type bound

Let $\mathcal{D}$ denote a tempered distribution and $\Psi$ a stationary kernel. Also, define $\hat{\mathcal{D}}(\omega) := \mathcal{D}_x e^{-i\langle\omega,\hat{x}\rangle}$.

**Proposition H.1.** *Let $h$ be a symmetric function such that for some $s \in (0,1]$, $h \in \mathcal{K}_{\Psi^{(s)}}$ and $\mathcal{D}_x h(\hat{x} - \cdot) \in \mathcal{K}_{\Psi^{(s)}}$. Then*

$$|\mathcal{D}_x h(\hat{x} - z)| \leq \|h\|_{\Psi^{(s)}} \left\|\mathcal{D}_x \Psi^{(s)}(\hat{x} - \cdot)\right\|_{\Psi^{(s)}}$$

*and for any $t \in (0,s)$ any function $\varrho(\omega)$,*

$$\left\|\mathcal{D}_x \Psi^{(s)}(\hat{x} - \cdot)\right\|_{\Psi^{(s)}}^{1-t} \leq \left(\left\|\varrho^{-1}\hat{\mathcal{D}}\right\|_{L^\infty} \left\|\varrho\hat{\Psi}^{t/2}\right\|_{L^2}\right)^{1-s} \|\mathcal{D}_x \Psi(\hat{x} - \cdot)\|_{\Psi}^{s-t}.$$

*Furthermore, if for some $c > 0$ and $r \in (0, s/2)$, $\hat{h} \leq c\,\hat{\Psi}^r$, then*

$$\|h\|_{\Psi^{(s)}} \leq \frac{c\left\|\Psi^{(r-s/2)}\right\|_{L^2}}{(2\pi)^{d/4}}.$$

**Proof** The first inequality follows from an application of Cauchy-Schwartz:

$$
\begin{aligned}
|\mathcal{D}_x h(\hat{x} - z)| &= |\langle h(\cdot - z), \mathcal{D}_x \Psi^{(s)}(\hat{x} - \cdot)\rangle_{\Psi^{(s)}}| \\
&\leq \|h(\cdot - z)\|_{\Psi^{(s)}} \left\|\mathcal{D}_x \Psi^{(s)}(\hat{x} - \cdot)\right\|_{\Psi^{(s)}} \\
&= \|h\|_{\Psi^{(s)}} \left\|\mathcal{D}_x \Psi^{(s)}(\hat{x} - \cdot)\right\|_{\Psi^{(s)}}.
\end{aligned}
$$

For the first norm, we have

$$
\begin{aligned}
\|h\|_{\Phi^{(s)}}^2 &= (2\pi)^{-d/2} \int \frac{\hat{h}^2(\omega)}{\hat{\Phi}^s(\omega)}\,\mathrm{d}\omega \\
&\leq c^2 (2\pi)^{-d/2} \int \hat{\Phi}^{2r-s}(\omega)\,\mathrm{d}\omega \\
&= c^2 (2\pi)^{-d/2} \left\|\Psi^{(r-s/2)}\right\|_{L^2}^2.
\end{aligned}
$$

Note that by the convolution theorem $\mathscr{F}(\mathcal{D}_x \Psi^{(s)}(\hat{x} - \cdot))(\omega) = \hat{\mathcal{D}}(\omega)\hat{\Psi}^s(\omega)$. For the second norm, applying Jensen's inequality and Hölder's inequality yields

$$
\begin{aligned}
\left\|\mathcal{D}_x \Psi^{(s)}(\hat{x} - \cdot)\right\|_{\Psi^{(s)}}^2 &= (2\pi)^{-d/2} \int \frac{\hat{\Psi}(\omega)^{2s}|\hat{\mathcal{D}}(\omega)|^2}{\hat{\Psi}^s(\omega)}\,\mathrm{d}\omega \\
&= (2\pi)^{-d/2} \left(\int \hat{\Psi}^t|\hat{\mathcal{D}}|^2\right) \int \frac{\hat{\Psi}(\omega)^t|\hat{\mathcal{D}}(\omega)|^2}{\int \hat{\Psi}^t|\hat{\mathcal{D}}|^2} \hat{\Psi}(\omega)^{s-t}\,\mathrm{d}\omega \\
&\leq (2\pi)^{-d/2} \left(\int \hat{\Psi}^t|\hat{\mathcal{D}}|^2\right) \left(\int \frac{\hat{\Psi}(\omega)^t|\hat{\mathcal{D}}(\omega)|^2}{\int \hat{\Psi}^t|\hat{\mathcal{D}}|^2} \Psi(\omega)^{1-t}\,\mathrm{d}\omega\right)^{\frac{s-t}{1-t}} \\
&= \left(\int \hat{\Psi}^t|\hat{\mathcal{D}}|^2\right)^{\frac{1-s}{1-t}} \|\mathcal{D}_x \Psi(\hat{x} - \cdot)\|_{\Psi}^{2\frac{s-t}{1-t}} \\
&\leq \left(\left\||\varrho^{-1}\hat{\mathcal{D}}|^2\right\|_{L^\infty} \int \varrho^2\hat{\Psi}^t\right)^{\frac{1-s}{1-t}} \|\mathcal{D}_x \Psi(\hat{x} - \cdot)\|_{\Psi}^{2\frac{s-t}{1-t}} \\
&= \left(\left\|\varrho^{-1}\hat{\mathcal{D}}\right\|_{L^\infty}^2 \left\|\varrho\hat{\Psi}^{t/2}\right\|_{L^2}^2\right)^{\frac{1-s}{1-t}} \|\mathcal{D}_x \Psi(\hat{x} - \cdot)\|_{\Psi}^{2\frac{s-t}{1-t}}.
\end{aligned}
$$

$\square$

# I   Verifying Example 3.3: Tilted hyperbolic secant RΦSD properties

We verify each of the assumptions in turn. By construction or assumption each condition in Assumption A holds. Note in particular that $\Psi_{2a}^{\text{sech}} \in C^\infty$. Since $e^{-a|x_d|} \leq \text{sech}(ax_d) \leq 2e^{-a|x_d|}$, Assumption B holds with $\|\cdot\| = \|\cdot\|_1$, $f(R) = 2^d e^{-\sqrt{\frac{\pi}{2}}aR}$, and $\underline{c} = 2^{-d}$, and $s = 1$. In particular,

$$\partial_{x_d} \log \Psi_{2a}^{\text{sech}}(x) = \sqrt{2\pi}a \tanh(\sqrt{2\pi}ax_d) + \textstyle\sum_{d' \neq d}^D \log \text{sech}(\sqrt{2\pi}ax_{d'})$$
$$\leq (\sqrt{2\pi}a)(1 + \textstyle\sum_{d' \neq d}^D |x_{d'}|)$$
$$\leq (\sqrt{2\pi}a)(1 + \|x\|_1)$$

and using Proposition L.3 we have that

$$\Psi_a^{\text{sech}}(x - z) \leq e^{\sqrt{\frac{\pi}{2}}a\|x\|_1} \Psi_a^{\text{sech}}(z) \leq 2^d \Psi_a^{\text{sech}}(z)/\Psi_a^{\text{sech}}(x).$$

Assumption C holds with $\overline{\lambda} = 1$ since for any $\lambda \in (0,1)$, it follow from Proposition L.2 that

$$\widehat{f}_j/\hat{\Phi}_j^{\lambda/2} = \hat{\Psi}_{2a}^{\text{sech}}/(\hat{\Psi}_a^{\text{sech}})^{\lambda/2} \leq 2^{d/2}(\hat{\Psi}_{2a}^{\text{sech}})^{1-\lambda} \in L^2.$$

The first part of Assumption D holds as well since by (6), $\omega_d^2 \hat{\Psi}_a^{\text{sech}}(\omega) = a^{-D} \omega_d^2 \Psi_{1/a}^{\text{sech}}(\omega) \in L^1$.

Finally, to verify the second part of Assumption D, we first note that since $r = 2$, $t = \infty$. The assumption holds since by Proposition L.2, $\hat{\Psi}_a^{\text{sech}}(\omega)/\hat{\Psi}_{2a}^{\text{sech}}(\omega)^2 \leq 1$.

# J   Verifying Example 3.4: IMQ RΦSD properties

We verify each of the assumptions in turn. By construction or assumption each condition in Assumption A holds. Note in particular that $\Psi_{c',\beta'}^{\text{IMQ}} \in C^\infty$. Assumption B holds with $\|\cdot\| = \|\cdot\|_2$, $f(R) = ((c')^2 + R^2)^{\beta'}$, $\underline{c} = 1$, and $s = 0$. In particular,

$$|\partial_{x_d} \log \Psi_{c',\beta'}^{\text{IMQ}}(x)| \leq -\frac{2\beta'|x_d|}{(c')^2 + \|x\|_2^2} \leq -2\beta'$$

and

$$\frac{\Psi_{c',\beta'}^{\text{IMQ}}(x - z)}{\Psi_{c',\beta'}^{\text{IMQ}}(z)} = \left(\frac{(c')^2 + \|x - z\|_2^2}{(c')^2 + \|z\|_2^2}\right)^{-\beta'}$$
$$\leq \left(\frac{(c')^2 + 2\|z\|_2^2 + 2\|x\|_2^2}{(c')^2 + \|z\|_2^2}\right)^{-\beta'}$$
$$\leq \left(2 + 2\|x\|_2^2/(c')^2\right)^{-\beta'}$$
$$= 2^{-\beta} \Psi_{c',\beta'}^{\text{IMQ}}(x)^{-1}.$$

By Wendland [29, Theorem 8.15], $\Psi_{c,\beta}^{\text{IMQ}}$ has generalized Fourier transform

$$\widehat{\Psi_{c,\beta}^{\text{IMQ}}}(\omega) = \frac{2^{1+\beta}}{\Gamma(-\beta)}\left(\frac{\|\omega\|_2}{c}\right)^{-\beta-D/2} K_{\beta+D/2}(c\|\omega\|_2),$$

where $K_v(z)$ is the modified Bessel function of the third kind. We write $a(\ell) \overset{.}{\sim} b(\ell)$ to denote asymptotic equivalence up to a constant: $\lim_\ell a(\ell)/b(\ell) = c$ for some $c \in (0, \infty)$. Asymptotically [1, eq. 10.25.3],

$$\hat{\Psi}_{c,\beta}^{\text{IMQ}}(\omega) \overset{.}{\sim} \|\omega\|_2^{-\beta-D/2-1/2} e^{-c\|\omega\|_2}, \qquad\qquad \|\omega\|_2 \to \infty \quad \text{and}$$
$$\hat{\Psi}_{c,\beta}^{\text{IMQ}}(\omega) \overset{.}{\sim} \|\omega\|_2^{-(\beta+D/2)-|\beta+D/2|} = \|\omega\|_2^{-(2\beta+D)_+} \qquad \|\omega\|_2 \to 0.$$

Assumption C holds since for any $\lambda \in (0, \overline{\lambda})$,

$$\hat{\Psi}^{\mathrm{IMQ}}_{c',\beta'}/(\hat{\Psi}^{\mathrm{IMQ}}_{c,\beta})^{\lambda/2} \sim \|\omega\|_2^{-(\beta'+D/2-1/2)+(\beta+D/2-1/2)\lambda/2} e^{(-c'+c\lambda/2)\|\omega\|_2}, \quad \|\omega\|_2 \to \infty \quad \text{and}$$

$$\sim \|\omega\|_2^{\lambda(2\beta+D)_+/2-(2\beta'+D)_+} = \|\omega\|_2^{\lambda(2\beta+D)/2} \qquad \|\omega\|_2 \to 0,$$

so $\hat{\Psi}^{\mathrm{IMQ}}_{c',\beta'}/(\hat{\Psi}^{\mathrm{IMQ}}_{c,\beta})^{\lambda/2} \in L^2$ as long as $c' = c\overline{\lambda}/2 > c\lambda/2$ and $\lambda(2\beta+D) > -D$. The first condition holds by construction and second condition is always satisfied, since $2\beta + D \geq 0 > -D$.

The first part of Assumption D holds as well since $\hat{\Psi}^{\mathrm{IMQ}}_{c',\beta'}(\omega)$ decreases exponentially as $\|\omega\|_2 \to \infty$ and $\hat{\Psi}^{\mathrm{IMQ}}_{c',\beta'}(\omega) \sim 1$ as $\|\omega\|_2 \to 0$, so $\omega_d^2 \hat{\Psi}^{\mathrm{IMQ}}_{c',\beta'}(\omega)$ is integrable.

Finally, to verify the second part of Assumption D we first note that $t = r/(2-r) = -D/(D+4\beta'\underline{\xi})$. Thus,

$$\hat{\Psi}^{\mathrm{IMQ}}_{c,\beta}/(\hat{\Psi}^{\mathrm{IMQ}}_{c',\beta'})^2 \mathrel{\dot\sim} \|\omega\|_2^{-2(\beta+D/2-1/2)/2+2(\beta'+D/2-1/2))} e^{2(-c/2+c')\|\omega\|_2}, \quad \|\omega\|_2 \to \infty \quad \text{and}$$

$$\mathrel{\dot\sim} \|\omega\|_2^{2(2\beta'+D)_+-(2\beta+D)_+} = \|\omega\|_2^{-(2\beta+D)} \qquad \|\omega\|_2 \to 0,$$

so $\hat{\Psi}^{\mathrm{IMQ}}_{c,\beta}/(\hat{\Psi}^{\mathrm{IMQ}}_{c',\beta'})^2 \in L^t$ whenever $c/2 > c'$ and

$$\frac{D}{(D + 4\beta'\underline{\xi})}(2\beta + D) > -D \Leftrightarrow -\beta/(2\underline{\xi}) - D/(2\underline{\xi}) > \beta'.$$

Both these conditions hold by construction.

## K    Proofs of Proposition 4.1 and Theorem 4.3: Asymptotics of $\mathrm{R}\Phi\mathrm{SD}$

The proofs of Proposition 4.1 and Theorem 4.3 rely on the following asymptotic result.

**Theorem K.1.** *Let $\xi_i : \mathbb{R}^D \times \mathcal{Z} \to \mathbb{R}, i = 1, \ldots, I$, be a collection of functions; let $Z_{N,m} \overset{indep}{\sim} \nu_N$, where $\nu_N$ is a distribution on $\mathcal{Z}$; and let $X_n \overset{i.i.d.}{\sim} \mu$, where $\mu$ is absolutely continuous with respect to Lebesgue measure. Define the random variables $\xi_{N,nim} := \xi_i(X_n, Z_{N,m})$ and, for $r, s \geq 1$, the random variable*

$$F_{r,s,N} := \left( \sum_{i=1}^I \left( \sum_{m=1}^M \left| N^{-1} \sum_{n=1}^N \xi_{N,nim} \right|^r \right)^{s/r} \right)^{2/s}.$$

*Assume that for all $N \geq 1$, $i \in [I]$, and $m \in [M]$, $\xi_{N,1im}$ has a finite second moment that that $\Sigma_{im,i'm'} := \lim_{N \to \infty} \mathrm{Cov}(\xi_{N,im}, \xi_{N,i'm'}) < \infty$ exists for all $i, i \in [I]$ and $m, m' \in [M]$. Then the following statements hold.*

1. *If $\varrho_{N,im} := (\mu \times \nu_N)(\xi_i) = 0$ for all $i \in [N]$ then*

$$NF_{r,s,N} \overset{\mathcal{D}}{\Longrightarrow} \left( \sum_{i=1}^I \left( \sum_{m=1}^M |\zeta_{im}|^r \right)^{s/r} \right)^{2/s} \text{ as } N \to \infty, \qquad (5)$$

    *where $\zeta \sim \mathcal{N}(0, \Sigma)$.*

2. *If $\varrho_{N,im} \neq 0$ for some $i$ and $m$, then*

$$NF_{r,s,N} \overset{a.s.}{\to} \infty \text{ as } N \to \infty.$$

**Proof**    Let $V_{N,im} = N^{-1/2} \sum_{n=1}^N \xi_{N,nim}$. By assumption $\|\Sigma\| < \infty$. Hence, by the multivariate CLT,

$$V_N - N^{1/2}\varrho_N \overset{\mathcal{D}}{\Longrightarrow} \mathcal{N}(0, \Sigma).$$

Observe that $NF_{r,s,N} = (\sum_{i=1}^I (\sum_{m=1}^M |V_{N,im}|^r)^{s/r})^{2/s}$. Hence if $\varrho = 0$, (5) follows from the continuous mapping theorem.

Assume $\varrho_{N,ij} \neq 0$ for some $i$ and $j$ and all $N \geq 0$. By the strong law of large numbers, $N^{-1/2}V_N \overset{a.s.}{\to} \varrho_\infty$. Together with the continuous mapping theorem conclude that $F_{r,s,N} \overset{a.s.}{\to} c$ for

some $c > 0$. Hence $NF_{r,s,N} \overset{a.s.}{\to} \infty$. $\square$

When $r = s = 2$, the R$\Phi$SD is a degenerate $V$-statistic, and we recover its well-known distribution [24, Sec. 6.4, Thm. B] as a corollary. A similar result was used in Jitkrittum et al. [16] to construct the asymptotic null for the FSSD, which is degenerate $U$-statistic.

**Corollary K.2.** *Under the hypotheses of Theorem K.1(1),*

$$NF_{2,2,N} \overset{\mathcal{D}}{\implies} \sum_{i=1}^{I} \sum_{m=1}^{M} \lambda_{im} \omega_{im}^2 \text{ as } N \to \infty,$$

*where* $\lambda = \text{eigs}(\Sigma)$ *and* $\omega_{ij} \overset{i.i.d.}{\sim} \mathcal{N}(0,1)$.

To apply these results to R$\Phi$SDs, take $s = 2$ and apply Theorem K.1 with $I = D$, $\xi_{N,dm} = \xi_{r,N,dm}$. Under $H_0 : \mu = P$, $P(\xi_{r,N,dm}) = 0$ for all $d \in [D]$ and $m \in [M]$, so part 1 of Theorem K.1 holds. On the other hand, when $\mu \neq P$, there exists some $m$ and $d$ for which $\mu(\xi_{r,dm}) \neq 0$. Thus, under $H_1 : \mu \neq P$ part 2 of Theorem K.1 holds.

The proof of Theorem 4.3 is essentially identical to that of Jitkrittum et al. [16, Theorem 3].

## L  Hyperbolic secant properties

Recall that the hyperbolic secant function is given by $\text{sech}(a) = \frac{2}{e^a + e^{-a}}$. For $x \in \mathbb{R}^d$, define the hyperbolic secant kernel

$$\Psi_a^{\text{sech}}(x) := \text{sech}\left(\sqrt{\frac{\pi}{2}} ax\right) := \prod_{i=1}^{d} \text{sech}\left(\sqrt{\frac{\pi}{2}} ax_i\right).$$

It is a standard result that

$$\hat{\Psi}_a^{\text{sech}}(\omega) = a^{-D} \Psi_{1/a}^{\text{sech}}(\omega). \tag{6}$$

We can relate $\Psi_a^{\text{sech}}(x)^\xi$ to $\Psi_{a\xi}^{\text{sech}}(x)$, but to do so we will need the following standard result:

**Lemma L.1.** *For* $a, b \geq 0$ *and* $\xi \in (0,1]$,

$$\frac{a^\xi + b^\xi}{2^{1-\xi}} \leq (a+b)^\xi \leq a^\xi + b^\xi.$$

**Proof**  The lower bound follows from an application of Jensen's inequality and the upper bound follows from the concavity of $a \mapsto a^\xi$. $\square$

**Proposition L.2.** *For* $\xi \in (0,1]$,

$$\Psi_a^{\text{sech}}(x)^\xi \leq \Psi_a^{\text{sech}}(\xi x) = \Psi_{a\xi}^{\text{sech}}(x) \leq 2^{d(1-\xi)} \Psi_a^{\text{sech}}(x)^\xi$$
$$2^{-d(1-\xi)} \hat{\Psi}_{a/\xi}^{\text{sech}}(x) \leq \hat{\Psi}_a^{\text{sech}}(x)^\xi \leq \hat{\Psi}_{a/\xi}^{\text{sech}}(x).$$

*Thus,* $\Psi_{a/\xi}^{\text{sech}}$ *is equivalent to* $\left(\Psi_a^{\text{sech}}\right)^{(\xi)}$.

**Proof**  Apply Lemma L.1 and (6). $\square$

**Proposition L.3.** *For all* $x, y \in \mathbb{R}^d$ *and* $a > 0$,

$$\Psi_a^{\text{sech}}(x - z) \leq e^{\sqrt{\frac{\pi}{2}} a \|x\|_1} \Psi_a^{\text{sech}}(z).$$

**Proof**  Take $d = 1$ since the general case follows immediately. Without loss of generality assume that $x \geq 0$ and let $a' = \sqrt{\frac{\pi}{2}} a$. Then

$$\frac{\Psi_a^{\text{sech}}(x-z)}{\Psi_a^{\text{sech}}(z)} = \frac{e^{a'z} + e^{-a'z}}{e^{a'(x-z)} + e^{-a'(x-z)}} = \frac{e^{a'z} + e^{-a'z}}{e^{-a'z} + e^{2a'x}e^{a'z}} e^{a'x} \leq e^{a'x}.$$

$\square$

# M    Concentration inequalities

**Theorem M.1** (Chung and Lu [5, Theorem 2.9]). *Let $X_1, \ldots, X_m$ be independent random variables satisfying $X_i > -A$ for all $i = 1, \ldots, m$. Let $X := \sum_{i=1}^m X_i$ and $\overline{X^2} := \sum_{i=1}^m \mathbb{E}[X_i^2]$. Then for all $t > 0$,*

$$\mathbb{P}(X \leq \mathbb{E}[X] - t) \leq e^{-\frac{1}{2}t^2/(\overline{X^2} + At/3)}.$$

Let $\hat{X} := \frac{1}{m} \sum_{i=1}^m X_i$.

**Corollary M.2.** *Let $X_1, \ldots, X_m$ be i.i.d. nonnegative random variables with mean $\bar{X} := \mathbb{E}[X_1]$. Assume there exist $c > 0$ and $\gamma \in [0, 2]$ such that $\mathbb{E}[X_1^2] \leq c\bar{X}^{2-\gamma}$. If, for $\delta \in (0, 1)$ and $\varepsilon \in (0, 1)$,*

$$m \geq \frac{2c \log(1/\delta)}{\varepsilon^2} \bar{X}^{-\gamma},$$

*then with probability at least $1 - \delta$, $\hat{X} \geq (1 - \varepsilon)\bar{X}$.*

**Proof**    Applying Theorem M.1 with $t = m\varepsilon\bar{X}$ and $A = 0$ yields

$$\mathbb{P}(\hat{X} \leq (1 - \varepsilon)\bar{X}) \leq e^{-\frac{1}{2}\varepsilon^2 m\bar{X}^2/(c\mathbb{E}[X_1^2])} \leq e^{-\frac{1}{2c}\varepsilon^2 m\bar{X}^\gamma}.$$

Upper bounding the right hand side by $\delta$ and solving for $m$ yields the result.    □

**Corollary M.3.** *Let $X_1, \ldots, X_m$ be i.i.d. nonnegative random variables with mean $\bar{X} := \mathbb{E}[X_1]$. Assume there exists $c > 0$ and $\gamma \in [0, 2]$ such that $\mathbb{E}[X_1^2] \leq c\bar{X}^{2-\gamma}$. Let $\epsilon' = |X^* - \bar{X}|$ and assume $\epsilon' \leq \eta X^*$ for some $\eta \in (0, 1)$. If, for $\delta \in (0, 1)$,*

$$m \geq \frac{2c \log(1/\delta)}{\varepsilon^2} \bar{X}^{-\gamma},$$

*then with probability at least $1 - \delta$, $\hat{X} \geq (1 - \varepsilon)X^*$. In particular, if $\epsilon' \leq \frac{\sigma X^*}{\sqrt{n}}$ and $X^* \geq \frac{\sigma^2}{\eta^2 n}$, then with probability at least $1 - \delta$, $\hat{X} \geq (1 - \varepsilon)X^*$ as long as*

$$m \geq \frac{2c(1-\eta)^2 \eta^{2\gamma}}{\varepsilon^2 \sigma^{2\gamma} \log(1/\delta)} n^\gamma.$$

**Proof**    Apply Corollary M.2 with $\frac{\varepsilon X^*}{\bar{X}}$ in place of $\varepsilon$.    □

**Example M.1.** If we take $\gamma = 1/4$ and $\eta = \varepsilon = 1/2$, then $X^* \geq \frac{4\sigma^2}{n}$ and $m \geq \frac{\sqrt{2}c \log(1/\delta)}{\sigma^{1/2}} n^{1/4}$ guarantees that $\hat{X} \geq \frac{1}{2}X^*$ with probability at least $1 - \delta$.