[Reviews · NeurIPS 2018]

Reviewer 1



Stein discrepancies have become a popular diagnosis tool emerging over the last few years for MCMC diagnosis. Most convergence tests such as asymptotic variance diagnostics, trace plots and so on work well in settings of no bias. However, for approximate MCMC with subset posteriors bias is introduced and stein discrepancies are a tool that allow us to confirm convergence. Over the past number of years research in this direction has proven to be generally quite impactful. Specifically in this paper, the authors address the complexity challenge associated with SD methods while avoiding a loss in quality of test. The authors introduce feature-SDs both in terms of construction and in terms of provable determination of convergence. Special cases of feature-SD are shown to equivalent to kernel stein discrepancies (KSDs), RFF approximations to KSDs and random finite set SDs (FSSDs), hence linking the work to broader research in the area. The assumptions required in the selection of feature functions, namely smoothness and decay rate less than Gaussian, seem reasonable in most practical settings. In the experimentation section, it appears only artificial data / problems are considered and hence act as a sketch of validation of the proposed method rather than the impact of the method in applications. This paper acts as a theoretical contribution and I believe that the experiments are sufficient for this purpose. I would say that the paper assumes the reader is very well versed in the stein discrepancies litterature and brushes over a more detailed introduction. However, while the paper may lose some non-expert readers, it is rich with content and it is unclear that the authors would have space in the 8 page limit to go into any more detail. Other notes: While the paper is relatively dense to read through, I have found no obvious mistakes in terms of the equations presented and spelling/grammar. Under many of the theorems the authors use short examples which clarify the above. This is extremely useful in order to follow the usefulness of each point and clarify the statements.

Reviewer 2



Edit: I increased the score because the authors answered my question about the derivation of sample complexity of M and the experimental verification and I'm satisfied with it. Summary The paper proposed a novel discrepancies between probabilistic distributions called \Phi Stein discrepancies (\PhiSD) and random \Phi Stein discrepancies (R\PhiSD) for the fast computation of Stein discrepancies. Existing kernel Stein discrepancies (KSD) requires O(N^2) computation that is time-consuming when N becomes large. To reduce the computation, the paper first proposed \PhiSD that is the generalization of KSD to Lr norm. And then replace the computation of Lr norm with importance sampling to construct R\PhiSD. They first prove that the proposed \PhiSD can upper bound the KSD for some kernel and then introduce \PhiSD for the kernel with enough non-convergence detecting power. Then they prove that R\PhiSD with M=\Omega(N^(\gammar/2))<=\Omega(N) samples can upper bound \PhiSD with high probability. Thus, we can use R\PhiSD to test the goodness of fit. Experimental results using synthesized data demonstrate that R\PhiSD with IMQ kernel shows good empirical performance. Qualitative Assessment The idea that generalization of KSD to Lr norm combined with importance sampling seems interesting. The proposed L1 IMQ is fast to compute and show good performance for Gaussian distribution. However, there seems some types and lack of explanations. Also, I have some question about main results. As for Proposition 1, the proof should be more detailed because this is one of the main results. I think the proof use the fact that F and QT are commutative. It is more understandable to write so. In the proof, do J mean D? Also, it should be noted that \rho \in Lt. In Proposition 3.4, I suspect that the righthand should be (1-\epsilon)^{2/r}\PhiSD^2 because R\PhiSD^2 is sum of 2/r power of w_d. Also, how is M=\Omega(N^{\gammar/2} derived? I think the author assumed that E[Y_d]=\Omega(N^-1), but I think this is not proved in the paper. As for the experiment, the proposed L1 IMQ requires much importance sample than L2 SechExp for Gaussian mixture in Fig 2. Also, the differences of discrepancy measures are much smaller than those of other methods. Though L1 IMQ show good performance for Gaussian dist in Figure 3, it is questionable if L1 IMQ is effective for complex distributions. I suggest to match the experimental setting to the previous works such as the paper of Kernel Stein Discrepancy. The paper will be more readable if ‘Proposition’ and ‘Assumption’ is written in bold. Also, the definition of KSD should be written in main section rather than in special cases. Overall, the idea seems novel and promising, but the theoretical and experimental explanations are not convincing enough.

Reviewer 3



The authors have addressed most of my concerns, and I maintain my overall evaluation. ---------- Based on recent work in graph Stein discrepancy and kernel Stein discrepancy (KSD), this paper proposes feature Stein discrepancies (\PhiSDs) as a new family of discrepancy measures that can be efficiently approximated using importance sampling (termed R\PhiSDs) which can be computed in near-linear time. By finding suitable KSDs that are upper-bounded by \PhiSDs, the paper shows that \PhiSDs provably determine the convergence of a sample to its target. The application of R\PhiSD to goodness-of-fit testing is discussed, and its asymptotic distribution is derived. Experiments are conducted on sampler selection in approximate inference as well as goodness-of-fit testing. Overall, the writing of the paper is fairly clear and the mathematical development quite rigorous, although at times it would help to provide some more intuition/motivation for the definitions/assumptions being introduced (such as the name “tilted base kernel” and the proposed hyperbolic secant kernel). I believe that this work is a nice addition to the existing literature on Stein discrepancies and goodness-of-fit testing. In particular, it is interesting to see that \PhiSDs include KSD and the finite-set Stein discrepancy (FSSD) as special cases. More specifically, I have the following specific questions/comments: - Since it is not a commonly used kernel function, what is the intuition and motivation for considering the hyperbolic secant kernel? Is it merely for mathematical convenience for verifying the various assumptions or are there certain application scenarios in which it would arise as a natural choice? - If one wishes to develop another instance of R\PhiSD, rather than checking all of Assumptions 3.1 through 3.6, is there a more concise but possibly slightly stronger set of conditions that could be more intuitive or easier to verify in practice and guide the choice of the various quantities in the definition of R\PhiSD? - While the authors have given two examples of R\PhiSDs, is it possible to provide a general characterization of how the choice of r affects the properties of the resulting R\PhiSD? Regarding the experiments: - In Figure 3(b) and (c), the behavior of L1-IMQ is quite strange, do the authors have any intuition/conjecture behind its counterintuitive behavior as dimension increases? Is this behavior consistent as one increase the number of features used? - For the computational complexity experiment, while the KSD test serves as a simple sanity check, I would be more interested in seeing how R\PhiSD compares with FSSD, since both should run in near-linear time, and it’s unclear which would be more efficient in practice. - For the goodness-of-fit testing experiments, it would be interesting and (more informative) to perform experiments on more complicated distributions (in addition to the standard Gaussian), such as those with multiple modes (e.g., Gaussian mixture models). In particular, should one expect better performance of \RPhiSD over KSD and KSSD in the presence of multi-modality?